# Flexible and Efficient Grammar-Constrained Decoding

**Kanghee Park** [1]  **Timothy Zhou** [1]  **Loris D'Antoni** [1]

## Abstract

Large Language Models (LLMs) are often asked to generate structured outputs that obey precise syntactic rules, such as code snippets or formatted data. Grammar-constrained decoding (GCD) can guarantee that LLM outputs matches such rules by masking out tokens that will provably lead to outputs that do not belong to a specified context-free grammar (CFG). To guarantee soundness, GCD algorithms have to compute how a given LLM subword tokenizer can "align" with the tokens used by a given context-free grammar and compute token masks based on this information. Doing so efficiently is challenging and existing GCD algorithms require tens of minutes to pre-process common grammars. We present a new GCD algorithm together with an implementation that offers 17.71x faster offline preprocessing than existing approaches while preserving state-of-the-art efficiency in online mask computation.

## 1. Introduction

Constrained decoding guides the output of Large Language Models (LLMs) by greedily enforcing user-given constraints in highly structured settings. *Grammar-constrained decoding* (GCD) (Geng et al., 2023) refers to the specific case where the constraint is given as a formal grammar that the LLM's output must conform to. This is done by applying parsing algorithms to build an automaton that interfaces with the LLM's decoding algorithm to mask away *all tokens that will provably lead to outputs not in the grammar*. For example, GCD can be used to ensure that an LLM generates programs that only use a specific set of function names.

Parsing algorithms for Context-Free Grammars (CFG) achieve efficiency by processing input strings in two phases. Terminals—e.g., variable names or string literals—are rec-

ognized by a lexer in a preprocessing phase, while the grammatical structure of how terminals can be arranged—e.g., that the body of a loop should be a sequence of statements—is enforced by an automaton operating over lexed terminals. A key challenge with implementing GCD algorithms is that *the tokens used by subword tokenizers in LLMs do not align with the terminals used by parsers*.

Because of this misalignment, GCD approaches either incur high *online token-masking overhead (>1 second per token)*, or high *offline preprocessing costs (>30 minutes)* to precompute a lookup table that relates LLM tokens to terminals (Beurer-Kellner et al., 2024; Ugare et al., 2024). Thus, existing GCD algorithms are impractical for domains where the constraining grammar frequently changes. Examples of such domains include *program synthesis* (Alur et al., 2019), where a grammar is provided for every program one may try to synthesize, and *grammar prompting* (Wang et al., 2024), where grammars are predicted from a given prompt to then guide the LLM towards a particular output structure.

In this paper, we introduce a new approach for grammar-constrained decoding that is both *flexible*—i.e., handles diverse grammars without incurring prohibitive offline preprocessing costs—and *efficient*—i.e., maintains state-of-the-art efficiency in online token masking. The key innovation is a combined analysis of the LLM token vocabulary and set of CFG terminals. The analysis efficiently precomputes a lexer-state-dependent mapping between sequences of CFG tokens and individual LLM tokens. This precomputation allows the decoder to efficiently identify valid LLM tokens at decoding time while avoiding wasteful processing of token combinations that are not *realizable* within the given LLM vocabulary.

We implement our approach in a tool, GREATGRAMMA, which compares favorably to existing GCD approaches. GREATGRAMMA achieves an average 17.71x speedup in offline preprocessing compared to related approaches (Ugare et al., 2024) while maintaining state-of-the-art online masking efficiency (5-32ms per token). Furthermore, our evaluation reveals that GCD implementations from existing published papers contain soundness bugs (e.g., they mask tokens that should not be masked or do not terminate when preprocessing simple grammars), whereas our approach lends itself to an elegant implementation consisting of simple

---

[1]Department of Computer Science and Engineering, UCSD, San Diego, USA. Correspondence to: Kanghee park <kap022@ucsd.edu>.

*Proceedings of the $42^{nd}$ International Conference on Machine Learning*, Vancouver, Canada. PMLR 267, 2025. Copyright 2025 by the author(s).

modules.

This paper makes two contributions:

- A flexible (low offline overhead) and efficient (low online token-masking overhead) algorithm for grammar-constrained decoding (Sec. 3-4).

- A tool implementing our algorithm, called GREAT-GRAMMA, that is up to 17.71x faster than other tools in offline preprocessing while retaining low online token-masking overhead (Sec. 5).

## 2. Approach Overview

In this section, we give some brief background about parsing (Sec. 2.1) and grammar-constrained decoding (Sec. 2.2), overview the high-level structure of our decoding approach (Sec. 2.3), and discuss our improvements over prior work.

### 2.1. Lexing and Parsing

While it is possible to build parsers that operate directly on input text, practical parsing tools first perform a preprocessing pass using a lexer for efficiency. A *lexer* is built from a set of regular expressions which define terminals (e.g. identifiers, keywords, or integer literals). Fig. 1a illustrates two regular expressions defining two terminals B (strings consisting of an a followed by a sequence of b's) and C (strings consisting of an a followed by a sequence of c's).

The lexer then takes as input a string and tokenizes it so that substrings matching regular expressions for terminals are grouped together. Here, our terminals consist of an a followed by either a sequence of b or c. For instance, the string "abaccab" lexes as ab acc ab.

The pass by the lexer ensures that the parser can be defined directly over terminals instead of individual characters. This separation of concerns avoids mixing character-level processing with higher-level grammar rules, making both components easier to implement and maintain. In particular, the grammar for the language can be defined at the *terminal* level. In our example, the grammar shown in Fig. 1e accepts sequences of terminals of the form BCBCBC... (where B and C can be any strings matching those terminals).

One way of defining a parser is as an automaton: it takes as input a sequence of terminals and either accepts or rejects the sequence. To handle all context-free grammars the automaton needs to be a pushdown automaton (PDA) that is equipped with a stack. In our example, the simple stack-free automaton shown in Fig. 1f is enough to describe all valid sequences of terminals.

### 2.2. Using Parsing for Constrained Decoding

We recall the general structure of a constrained decoder. When given a sequence of tokens $t_{1:k}$, CONSTRAINEDDE-CODING (Alg. 1 in Appendix A) uses a checker $C$ during the decoding process to mask out any next token $t_{k+1}$ that would result in a prefix $t_{1:k+1}$ for which no possible completion satisfies the constraint. Specifically, given a prefix $t_{1:k}$ and vocabulary $\mathcal{V} \subseteq \Sigma^+$, a checker $C$ computes a Boolean mask $C(t_{1:k}; \mathcal{V}) = m \in \{0, 1\}^{|\mathcal{V}|}$, where a value of 1 denotes a viable token (i.e., one for which there exists a sentence completion that can lead to constraint satisfaction), and 0 denotes an invalid one (i.e., one for which no sentence completion can lead to constraint satisfaction). The mask is then applied to the logits (Deutsch et al., 2019) produced by the language model to exclude invalid tokens from consideration.

Going back to grammars, a parser either accepts or rejects a string. A simple extension is to make this process online—the parser rejects as soon as it consumes a token that will ensure the input string fails to parse. Intuitively, the key idea behind *grammar-constrained decoding* is that the parser primitive for checking if a token will be allowed or not by the parser can be used to build the checker $C$.

However, there is a key problem with using a parser as a checker for constrained decoding: a parser reads *language* level tokens (which this paper calls *terminals* to avoid confusion), while an LLM outputs tokens according to its subword vocabulary. The LLM tokens may span multiple or only fragments of language terminals, complicating the decoding process. This problem is known as *token misalignment* (Poesia et al., 2022), and is the chief source of complexity for GCD implementations. For example some of the LLM tokens in Fig. 1c correspond to terminals whereas others can span multiple terminals.

### 2.3. Our Approach

Fig. 1 illustrates the overall structure of our approach, which is based on a data structure we introduce called a *token spanner table*. Intuitively, the token spanner table stores what sequences of terminals could be emitted by the lexer if it reads an LLM token from a given state.

Our GCD approach is split into two parts: an offline preprocessing phase in which the table is constructed and an online phase where the table is then used to generate token masks during decoding.

The offline portion of our algorithm takes two inputs: the LLM vocabulary $\mathcal{V}$ (Fig. 1c) and the definitions (as regexes) for terminals (Fig. 1a). It then constructs a finite state automaton (FSA) with a set of states $Q$ representing the lexer (Fig. 1b). For example, state $q_2$ represents a state where the lexer has already read the substring ab and can (optionally)

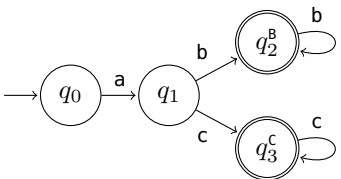

(a) Terminals and regular expressions.

"a", "b", "c", "ab", "ac", "aba"

(c) Subword vocabulary for the LLM.

$S := \mathtt{BC} \mid \mathtt{BC}S$

(e) Grammar $\mathcal{G}_{\mathtt{BC}}$.

(b) Lexing automaton obtained from terminal definitions in Fig. 1a.

|       | a      | b | c | ab  | ac  | aba      |
|-------|--------|---|---|-----|-----|----------|
| $q_0$ | B,C    | ∅ | ∅ | B   | C   | BB, BC   |
| $q_1$ | ∅      | B | C | ∅   | ∅   | ∅        |
| $q_2$ | BB, BC | B | ∅ | BB  | BC  | BBB, BBC |
| $q_3$ | CB, CC | ∅ | C | CB  | CC  | CBB, CBC |

(d) Token spanner table computed from lexing automaton (Fig. 1b) and subword vocabulary (Fig. 1c).

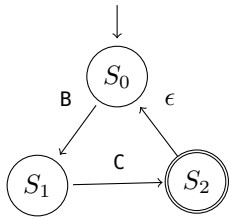

(f) Automaton for the grammar in Fig. 1e.

Figure 1: Illustrative example of the approach implemented in GREATGRAMMA.

read more b's to emit a B terminal. We use this automaton together with the LLM's vocabulary to build the token spanner table (Fig. 1d). The keys of the token spanner table are pairs $(q, v)$ where $q \in Q$ is a state of the lexer automaton and $v \in \mathcal{V}$ is an LLM token. The $(q, v)$ entry of the token spanner table contains the set of sequences of terminals that can eventually be produced if the lexer is in state $q$ and is fed in one additional LLM token $v$. For example, given state $q_2$ and LLM token aba we can either produce a sequence of terminals BBB or BBC—this is because we are in a lexer state corresponding to having started a B token, lex a complete B token from ab, and then start an arbitrary new token.

In the online portion, our algorithm uses the token spanner table together with the parser to construct masks stating which LLM tokens are valid at each step. During decoding, the algorithm tracks the state of the lexer (Fig. 1b) and the state of the parser PDA (Fig. 1f) on the prefix the LLM has generated so far. The algorithm then analyzes the state the parser is in to determine the possible sequences of terminal the parser could consume next. Then, the algorithms consult the token spanner table using these sequences and the current state of the lexer to determine what LLM tokens should be masked and kept.

## 3. Offline Token Preprocessing

Our approach starts by preprocessing the lexer to efficiently construct a lookup table that relates LLM tokens to terminal sequences (Sec. 3.1) and vice versa (Sec. 3.2). The preprocessed lexer is then used to analyze the parser to determine what terminal sequences are actually possible sequences in the grammar (Sec. 3.3).

### 3.1. Lexer Preprocessing

Let $\Sigma$ denote the set of string characters, $\Sigma^*$ denote the set of strings over this alphabet, and $\Gamma$ denote the set of terminals

(i.e., grammar-level tokens). A *lexer* is a function Lex that given an input string $w \in \Sigma^*$, returns a pair $(T_1 \ldots T_k, w_r)$, where $T_1 \ldots T_k \in \Gamma^*$ is a sequence of terminals and $w_r \in \Sigma^*$ is the suffix of $w$ that has not been lexed (i.e., mapped to language terminals) yet.

Typically lexers resolve ambiguity by making some simplifying assumptions that also help improve efficiency and avoid backtracking. We use the same assumptions and describe them next.

**Maximal Munch Principle and Lookahead**  Consider a language that contains two different tokens for the increment operator ++ and the addition operator +. Although the input string ++ could be tokenized as two separate + addition tokens, in practice lexers prioritize the longer valid token to resolve ambiguity (and which usually captures the intended syntax of the programming language). This behavior is called the *maximal munch principle*: the lexer matches the longest possible substring starting at the current position that aligns with a defined token pattern.

Under a strict interpretation of the maximal munch principle, if the lexer reaches the end of the input stream while processing a partial triple-quoted Python string """a", the lexer should tokenize the input as two strings "" and "a". However, supporting such cases would require either waiting until the end of the input string to produce any tokens or allowing backtracking. As such, in practice many lexers (including Python's) will raise an error and stop lexing if the scanned prefix cannot be tokenized as a single terminal. This greedy behavior disallows some strings, but guarantees that the lexer can resolve all tokenization ambiguities by inspecting only the next character at each step.

**Definition 3.1** (1-lookahead). A lexer Lex is *1-lookahead* if for every string $w \in \Sigma^*$ and valid continuation $c \in \Sigma$ of $w$, whenever $\text{Lex}(w) = (T_1 \ldots T_k, r)$ then $\text{Lex}(wc)$ is either $(T_1 \ldots T_k T_{k+1}, c)$ for some $T_{k+1} \in \Gamma$ or $(T_1 \ldots T_k, rc)$.

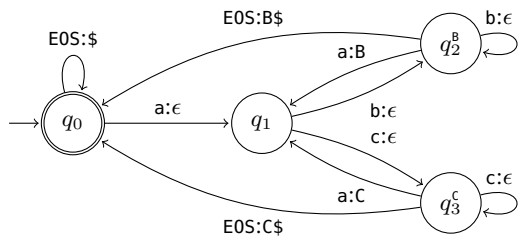

Figure 2: A lexing transducer $\mathcal{T}_{\mathcal{A}}$ derived from FSA $\mathcal{A}$ in Fig. 1.

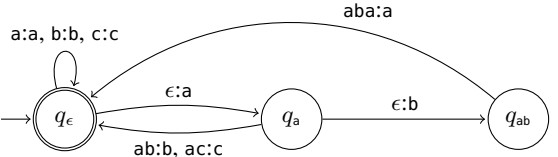

Figure 3: Detokenizing transducer for vocabulary $\mathcal{V} = \{a, b, c, ab, ac, aba\}$.

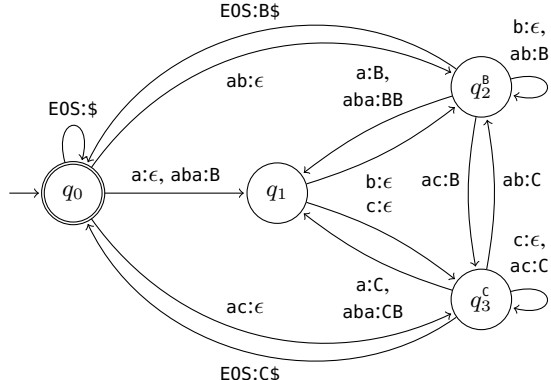

Figure 4: A determinized token-level lexing transducer $\mathcal{T}_{\mathcal{A} \circ \mathcal{V}}$, which is formed by composing $\mathcal{T}_{\mathcal{V}}$ from Fig. 3 and $\mathcal{T}_{\mathcal{A}}$ from Fig. 2.

Terminals are specified as a set of regular expressions. It is oftentimes convenient to work with a *lexing automaton*, which is the finite state automaton (FSA) that accepts strings matching any terminal definition (McNaughton & Yamada, 1960). We refer the reader to Sec. B.1 for formal definitions of the semantics of FSA, but recall that an FSA is a tuple $\mathcal{A} = (\Sigma, Q, q_0, \delta, F)$ where $\Sigma$ is the alphabet, $Q$ is the set of states with initial state $q_0 \in Q$, $\delta$ contains transitions of the form $q \xrightarrow{c} q'$, and $F \subseteq Q$ is the set of final states.

**Lexing Transducer** A 1-lookahead maximal munch lexer can be defined from a lexing automaton as follows: The input is processed character-by-character by transitioning through the FSA's states. When no valid transition exists for the next character $c$, the lexer checks whether the current state corresponds to a valid language token. If it does and the tokenizer has at this point produced a pair $(T_1 \ldots T_k, w_r)$, the not-yet tokenized string $w_r$ is tokenized with token $T_{k+1}$ corresponding to the reached state, the FSA is reset to its initial state $q_0$ (and the tokenizer state $(T_1 \ldots T_k T_{k+1}, \varepsilon)$ with the empty string $\varepsilon$), and the character $c$ is consumed as the starting character of a new token $q_0$. If the current state does not correspond to a valid token or if $c$ cannot be consumed at $q_0$, then $c$ is invalid. Invalid characters inform what tokens should be masked during constrained decoding.

This process can be formalized as a finite-state transducer (FST), an extension of a finite-state automaton that can produce output terminals when reading characters. Given the original lexing automaton $\mathcal{A}$ representing valid tokens, we write $\mathcal{T}_{\mathcal{A}}$ to denote the *lexing transducer*, the FST corresponding to $\mathcal{A}$. The construction of $\mathcal{T}_{\mathcal{A}}$ from $\mathcal{A}$ is formalized in Alg. 2 in Appendix A, but at a high level the process simply adds transitions to handle cases where no valid transition exists for the current character $c$, outputting terminals and exiting final states. Fig. 2 shows the lexing transducer derived from the FSA in Fig. 1.

**LLM Token to Terminals** Processing LLM tokens character-by-character with the lexing transducer at runtime would incur significant overhead. To address this problem, we instead construct a token-to-terminal transducer by composing the character-to-terminal lexing transducer with the detokenizing transducer introduced by Koo et al. (2024). A

detokenizing transducer simply maps LLM tokens to their corresponding sequence of text characters (i.e., the input alphabet is $\mathcal{V}$ and the output alphabet is $\Sigma$). A detokenizing transducer is nondeterministic and can contain $\varepsilon$-transitions that produce outputs without consuming inputs.

Fig. 3 illustrates the detokenizing transducer $\mathcal{T}_{\mathcal{V}}$ derived from the vocabulary $\mathcal{V} = \{a, b, c, ab, ac, aba\}$ in Fig. 1. Note that as an optimization common prefixes of tokens form a trie-like structure (e.g., state $q_a$ denotes the state reached when producing the first $a$ for all tokens that start with that character), reducing computational overhead by reusing shared prefix computation. Fig. 4 depicts the combined token-level lexing transducer $\mathcal{T}_{\mathcal{A} \circ \mathcal{V}} = \mathcal{T}_{\mathcal{A}} \circ \mathcal{T}_{\mathcal{V}}$, which is the determinized FST capturing the sequential functional composition of the two transducers—i.e., the output of $\mathcal{T}_{\mathcal{V}}$ is fed as input to $\mathcal{T}_{\mathcal{A}}$. This token-to-terminal transducer enables efficient lookup of valid next tokens and produced terminal symbols in each state.

### 3.2. Realizable Terminal Sequences

Now that we have defined the formal machinery behind lexing, we are ready to explain how LLM tokens can be mapped to possible sequences of terminals.

When the transducer $\mathcal{T}_{\mathcal{A} \circ \mathcal{V}}$ in Fig. 4 consumes the LLM token $aba$ from the initial state $q_0$, it produces the terminal $B$ and moves to state $q_1$. If we inspect the grammar $\mathcal{G}_{BC}$ in

---

**Algorithm 4** BUILDINVERSETOKENSPANNERTABLE

---

**Input:** Combined FST $\mathcal{T}_{\mathcal{A} \circ \mathcal{V}} = (\mathcal{V}, \Gamma, Q, q_0, \delta, F)$
**Output:** Realizable token sequences $Re_{\mathcal{A} \circ \mathcal{V}}$,
Inverse lookup table $T_{\text{inv}} : Re_{\mathcal{A} \circ \mathcal{V}} \times Q \to 2^{\mathcal{V}}$
$Re_{\mathcal{A} \circ \mathcal{V}} := \emptyset, T_{\text{inv}}(\cdot, \cdot) := \emptyset$
**for** $q \xrightarrow{t:T_1 \dots T_k} q' \in \delta$ **do**
$\quad$ **for** $T$ recognized at $q''$, that is reachable from $q'$ **do**
$\quad\quad Re_{\mathcal{A} \circ \mathcal{V}} := Re_{\mathcal{A} \circ \mathcal{V}} \cup \{T_1 \dots T_k T\}$
$\quad\quad T_{\text{inv}}(q, T_1 \dots T_k T) := T_{\text{inv}}(q, T_1 \dots T_k T) \cup \{t\}$
**return** $Re_{\mathcal{A} \circ \mathcal{V}}, T_{\text{inv}}$

---

Fig. 1e, we can deduce that the parser, which receives as input sequences of tokens, expects/requires the next language token to be C. Since $\mathcal{T}_{\mathcal{A} \circ \mathcal{V}}$ in state $q_1$ does not immediately produce any output when consuming b, the generated terminal sequence so far (i.e., B) is still a valid prefix according to the grammar $\mathcal{G}_{\text{BC}}$. However, after transitioning to $q_2$, no possible path can generate C next.

As illustrated by the above example, for each transition $q \xrightarrow{t:T_1 \dots T_k} q'$ in the lexing transducer $\mathcal{T}_{\mathcal{A} \circ \mathcal{V}}$, we should check whether there is a terminal $T$ such that (*i*) $T$ can be generated along some path from $q'$, and (*ii*) $T_1 \dots T_k T$ is accepted by the grammar. This observation leads to the following definition, which describes which terminal sequences need to be considered by the parser.

**Definition 3.2** (Producible Terminals). Given a state $q$, the set of producible terminals $Prod(q)$ is defined as the set

$$\{T_1 \in \Gamma \mid q \xrightarrow{w:T_1 \dots T_k}^{*} q' \in \delta^* \text{ for some } q' \in Q, w \in \Sigma^*\}$$

**Definition 3.3** (Realizable Terminal Sequences). Given a token vocabulary $\mathcal{V}$ and FSA $\mathcal{A}$, let $\delta$ be the set of transitions in the token-level lexing transducer $\mathcal{T}_{\mathcal{A} \circ \mathcal{V}}$. The set of *realizable terminal sequences* $Re_{\mathcal{A} \circ \mathcal{V}}$ is defined as

$$Re_{\mathcal{A} \circ \mathcal{V}} = \{T_1 \dots T_k T \mid$$
$$\exists q, q', t. \, q \xrightarrow{t:T_1 \dots T_k} q' \in \delta \text{ and } T \in Prod(q')\}.$$

Note that the LLM vocabulary $\mathcal{V}$ contains all characters in $\Sigma$, ensuring that $\Sigma^* = \mathcal{V}^*$. Therefore, any next terminal producible in $\mathcal{T}_{\mathcal{A}}$ is also producible in the combined transducer $\mathcal{T}_{\mathcal{A} \circ \mathcal{V}}$ and vice versa. This equivalence allows us to simplify producibility checks: instead of analyzing the large combined transducer $\mathcal{T}_{\mathcal{A} \circ \mathcal{V}}$, we need only compute reachability to accepting states within $\mathcal{T}_{\mathcal{A}}$ to determine producible next terminals.

**Inverse Token Spanner Table** Alg. 4 computes the set of realizable terminal sequences and constructs the key data structure we use to perform constrained decoding.

**Definition 3.4** (Inverse Token Spanner Table). Given a lexer state $q \in Q_{\mathcal{A} \circ \mathcal{V}}$ and $T_1 \dots T_k T \in Re_{\mathcal{A} \circ \mathcal{V}}$, the entry $T_{\text{inv}}(q, T_1 \dots T_k T)$ in the *inverse token spanner table* $T_{\text{inv}}$ is the set of tokens that generates $T_1 \dots T_k T$ from state $q$. Formally,

$$T_{\text{inv}}(q, T_1 \dots T_k T) =$$
$$\{t \mid q \xrightarrow{t:T_1 \dots T_k} q' \in \delta \text{ and } T \in Prod(q')\}$$

Note that, for a given state $q'$, the set of producible terminals $T$ in the lexing transducer can be precomputed using reachability algorithm (Floyd, 1962). This table allows a decoder to determine which LLM tokens can result in a given realizable terminal sequence $T_1 \dots T_k T$ being produced from a specific lexer state.

Going back to the example in Fig. 1, the table in Fig. 1d illustrates the terminal sequences produced by the lexing transducer for each pair of state and token. The set of all realizable terminal sequences, $Re_{\mathcal{A} \circ \mathcal{V}}$, is obtained by taking the union of all entries in this table.

Building on Fig. 1d, we can also derive an inverse token spanner table, $T_{\text{inv}}$. For a given state $q$ and terminal sequences $\alpha$, the table $T_{\text{inv}}$ provides the set of tokens that can generate the sequence $\alpha$ from the state $q$. For example, the terminal sequence BC is generated by the token aba in state $q_0$ (i.e., aba $\in T_{\text{inv}}(q_0, \text{BC})$). The same sequence is generated by the token ac in state $q_2$ (i.e., ac $\in T_{\text{inv}}(q_2, \text{BC})$).

### 3.3. Parser Preprocessing

We do not formally define context-free grammars (CFG) for brevity and refer the reader to Sec. B.2 for details. For the sake of this paper, the reader only needs to know that a CFG parser is typically formalized as a pushdown automaton (PDA), an extension of FSA with an execution stack (Schützenberger, 1963). The definition of a PDA is similar to that of an FSA, but transitions are also allowed to manipulate a stack over symbols $\Pi$ via push and pop operations. Therefore, in a PDA, a configuration after reading a sequence of input characters is a pair of automaton state $q \in Q$ and execution stack $\gamma \in \Pi^*$.

We refer the reader to Sec. B.4 for the formal definition of a PDA, but informally, each PDA transition $q \xrightarrow{c[\beta \to \beta']} q'$ can only be activated if the character being read is a $c$ and the top of the current stack configuration $\gamma$ matches to sequence of stack symbols $\beta \in \Pi^*$. If the transitions is activated, the current state becomes $q'$, and the top $\beta$ elements of the stack are replaced with new stack elements $\beta' \in \Pi^*$.

As with lexer preprocessing, our goal is to also preprocess the parser to avoid iterating over every terminal sequence generated by the lexer at runtime. The challenge with pushdown automata (PDAs), compared to lexers (NFAs), is their

requirement for a stack to parse context-free grammars. This reliance on the stack prevents us from fully determining at preprocessing time whether a given terminal sequence is acceptable from a particular state, as the stack's contents must be examined during execution. However, we can still identify many terminal sequences that will always be accepted or always rejected, independent of the current stack.

To compute which terminal sequences will always be accepted or always rejected, we directly compose the detokenizing transducer $Re_{\mathcal{A} \circ \mathcal{V}}$ (Definition 3.3), with the PDA $\mathcal{P}$ produced by a parser generator (and where transitions operate over single terminals) (Allauzen & Riley, 2012) to obtain a new pushdown automaton $\mathcal{P} \circ \mathcal{T}_{Re_{\mathcal{A} \circ \mathcal{V}}}$ where transitions operate over terminal sequences.

This last pushdown automaton can efficiently determine valid sequences of terminal symbols from each parser state. We note that preprocessing both the lexer and parser in tandem is a key feature that distinguishes our work from prior work (Beurer-Kellner et al., 2024; Ugare et al., 2024).

*Stack invariance* provides a sufficient condition for knowing when a sequence of terminals is accepted.

**Proposition 3.5** (Stack Invariance). *If a PDA $\mathcal{P}$ accepts an input sequence $w$ in state $q$ with stack configuration $\gamma$, then $w$ is also accepted in the same state $q$ when the stack configuration is $\gamma' \cdot \gamma$ for some $\gamma'$ (i.e., when $\gamma$ appears at the top of the stack with additional symbols beneath it).*

It follows that a terminal sequence accepted by the PDA starting with an empty stack configuration is accepted under any stack configuration.

The following proposition shows how to construct a stack-free finite-state automaton that overapproximates the set of sequences accepted by a PDA.

**Proposition 3.6** (Overapproximation via FSA). *Consider an FSA $\mathcal{A}_{\mathcal{P}}$ obtained by removing all stack operations from a PDA $\mathcal{P}$. If an input sequence $w$ is not accepted by $\mathcal{A}_{\mathcal{P}}$ in state $q$, then $w$ cannot be accepted by $\mathcal{P}$ in state $q$ with any stack configuration.*

Following Proposition 3.6, a terminal sequence is always rejected if it is rejected by the FSA obtained by removing all stack operations.

The above reasoning can be formalized by computing the set of *always-accepted tokens* $A$ and of *context-dependent terminal sequences* $D$. Given a lexer state $q^{\mathcal{A}}$ and a parser state $q^{\mathcal{P}}$, we denote by $A(q^{\mathcal{A}}, q^{\mathcal{P}})$ the set of tokens that are accepted regardless of the stack configuration $\gamma$, and by $D(q^{\mathcal{A}}, q^{\mathcal{P}})$ the set of terminal sequences that may be accepted by some configuration $\gamma$.

Alg. 5 in Appendix A describes how to preprocess a parser to build a table of always-accepted tokens $A$ and context-

---

**Algorithm 6** COMPUTEVALIDTOKENMASK

> **Input:** PDA $\mathcal{P}$, FSA $\mathcal{A}$, Inverse token spanner table $T_{\text{inv}}$
> Always-accepted token table $A$,
> Context-dependent sequence table $D$,
> Lexer state $q^{\mathcal{A}}$, Parser state $q^{\mathcal{P}}$, Stack configuration $\gamma$
> **Output:** Set of allowed tokens $\mathcal{V}_{allowed}$
> $\mathcal{V}_{allowed} := A(q^{\mathcal{A}}, q^{\mathcal{P}})$
> **for** $\alpha \in D(q^{\mathcal{A}}, q^{\mathcal{P}})$ **do**
>     **if** $\alpha$ is accepted by $\mathcal{P}$ in state $q^{\mathcal{P}}$ with stack $\gamma$ **then**
>         $\mathcal{V}_{allowed} := \mathcal{V}_{allowed} \cup T_{\text{inv}}(q^{\mathcal{A}}, \alpha)$
> **return** $\mathcal{V}_{allowed}$

---

dependent sequences $D$.

Note that one key source of efficiency resulting from our approach is that many transitions in the combined transducer $\mathcal{T}_{\mathcal{A} \circ \mathcal{V}}$ produce the same terminal sequence $T_1 \ldots T_k T$, making $|Re_{\mathcal{A} \circ \mathcal{V}}|$ smaller than $|\mathcal{V}|$ or $|\delta|$. Thus, the set of realizable terminal sequences $Re_{\mathcal{A} \circ \mathcal{V}}$ enables efficient precomputation of what sequences of terminals the parser should consider.

## 4. Online Token Masking

With a preprocessed inverse token spanner table $T_{\text{inv}}$, along with the table of always-accepted tokens $A$ and context-dependent sequences $D$, we are now ready to describe the online part of our grammar-constrained decoder (Alg. 6).

At each decoding step, Alg. 6 analyzes the lexer state $q^{\mathcal{A}}$, parser state $q^{\mathcal{P}}$, and the current stack configuration $\gamma$ to produce the set of exactly all LLM tokens $\mathcal{V}_{allowed}$ that can lead to a sequence accepted by the the grammar.

Although we construct the combined PDA $\mathcal{P} \circ \mathcal{T}_{Re_{\mathcal{A} \circ \mathcal{V}}}$ for preprocessing what terminal sequences a token can result in, using the same automaton for checking at decoding time whether LLM tokens can lead to valid parsing sequences would be inefficient. PDAs cannot always be determinized and one would have to compute reachability for nodes in $\mathcal{P} \circ \mathcal{T}_{Re_{\mathcal{A} \circ \mathcal{V}}}$ under different stack configurations to determine valid language token sequences. This computation would effectively involve testing all sequences in $Re_{\mathcal{A} \circ \mathcal{V}}$ at decoding time, making the preprocessing of the parser meaningless. Therefore, Alg. 6 uses the set of always-accepted tokens $A$ whenever possible to avoid traversing the PDA and only analyzes what terminal the PDA can accept at decoding time for context-dependent sequences described by $D$. The set of next legal tokens is then obtained by looking up, for each sequence of terminals accepted by the grammar, the set of tokens that can lead to that sequence. This is done by consulting the inverse token spanner table $T_{\text{inv}}$.

# 5. Experiments

We implemented our algorithms for computing token masks in a tool called GREATGRAMMA. A testament to the simplicity of our approach is that GREATGRAMMA is implemented in just *900 lines of Python code* built on top of the LALR parser provided by the LARK library (Shinan et al.). Specifically, we used LARK to parse regular expressions and context-free grammars, and used LARK's LALR parser to generate a parse table representing a deterministic PDA for parsing sequences of tokens in a given context-free grammar.

In this section, we evaluate GREATGRAMMA by answering the following two questions:

- **RQ1:** How does the *offline preprocessing overhead* of GREATGRAMMA compares to existing GCD approaches?

- **RQ2:** How does the *online per-token decoding overhead* of GREATGRAMMA compares to existing GCD approaches?

Because all GCD approaches in theory produce the same token masks and only differ in execution time, we do not need to evaluate GREATGRAMMA's performance on specific downstream tasks; the effectiveness of GCD has already been evaluated in prior work (Geng et al., 2023; Beurer-Kellner et al., 2024; Park et al., 2024).

**Models and Grammars.** We conduct our experiments using three different tokenizers: Llama (32K tokens), Llama-3 (128K), and Qwen-2 (151K). We consider simplified Go, Java, and Python grammars from prior work on GCD (Ugare et al., 2024). We choose these grammars for three reasons: they are large grammars (87-99 terminals, 109-145 nonterminals, 353-520 production rules, 7,441-9,319 bytes), they capture real programming languages that are used in existing applications of constrained decoding, and they illustrate well the trade-off between offline preprocessing and online masking times. While other smaller grammars appear in existing work on GCD, the tools we compare against all take slightly different grammar formats with various restrictions (e.g., no left recursion was allowed in XGRAMMAR), and we had to manually translate all grammars we considered between these formats to perform our evaluation (a very time-consuming task).

**Baselines.** We compare GREATGRAMMA against OUT-LINES (Willard & Louf, 2023), SYNCODE (Ugare et al., 2024), and XGRAMMAR (Dong et al., 2024), the three state-of-the-art GCD tools that can nominally handle grammars of practical sizes.

**Measures.** For each combination of tokenizer and grammar, we measured the time taken to preprocess the grammar for that tokenizer and the average time taken by each GCD implementation to produce a token at inference time. To fairly evaluate per-token computation time, we wanted to ensure that all three tools followed the same token selection process and recorded the time required to compute the mask at each step—i.e., the online overhead. We manually built 5 programs in each grammar and used them to decide what sequence of tokens the decoder was going to produce—i.e., for each example program, the decoder produced each token in the program in order and computed the token masks at each step. Following this setup, we could exactly measure the same online average per-token overhead across all GCD approaches.

**Results.** Tab. 1 reports the results. We do not include XGRAMMAR in Tab. 1 because it encountered errors during either preprocessing or decoding for all the grammars we considered—it failed to compile the Java grammar, and entered an infinite loop or encountered a segmentation fault during decoding for Python and Go grammars.

On average (geomean), GREATGRAMMA's *offline preprocessing* is 17.71x faster than SYNCODE, but 30.01x slower than OUTLINES.

For certain terminal sequences, SYNCODE incorrectly masked a valid token (we suspect the source of this imprecision is that SYNCODE only unrolls future terminal sequences up to a fixed bound $k$). When considering the remaining data points, GREATGRAMMA's *online masking* is on average (geomean) 2.73x faster than SYNCODE and 550.57x faster than OUTLINES.

**Discussion.** In summary, GREATGRAMMA emerges as the new state-of-the-art GCD approach. GREATGRAMMA is 17.71x faster at offline preprocessing than SYNCODE, the only other GCD tool with acceptable online overhead (**RQ1**), and exhibits the lowest online masking overhead than any other GCD tool (**RQ2**).

While OUTLINES has the lowest preprocessing offline overhead, its seconds-per-token online overhead does not meet the needs of most practical applications of LLMs. OUTLINES' online overhead is due to the fact that its CFG module verifies the acceptability of each token at runtime.

The improvement in offline pre-processing over SYNCODE is expected as our new approach targets the inefficiency of SYNCODE's algorithm for token alignment. The slight improvement in online token-mask computation over SYNCODE is likely due to how GREATGRAMMA only consults the PDA for context-dependent sequences, while SYNCODE does so for all terminal sequences up to a bound.

Table 1: Offline and online processing times. X denotes when SYNCODE masked incorrect tokens.

| Model | $|V|$ | Grammar | Offline preprocessing (**seconds**) | | | Online per-token overhead (**milliseconds**) | | |
|---|---|---|---|---|---|---|---|---|
| | | | OUTLINES | SYNCODE | GREATGRAMMA | OUTLINES | SYNCODE | GREATGRAMMA |
| Llama | 32,000 | Go | 2.51 | 556.10 | 24.76 | 3,213.05 | X | **5.08** |
| | | Java | 2.49 | 579.48 | 33.79 | 3,450.67 | 46.71 | **6.29** |
| | | Python | 4.89 | 768.33 | 35.40 | 3,245.10 | 62.40 | **7.11** |
| Llama-3 | 128,256 | Go | 2.43 | 2,212.95 | 106.07 | 12,974.43 | X | **21.11** |
| | | Java | 2.46 | 2,294.26 | 166.87 | 13,725.58 | 49.03 | **24.69** |
| | | Python | 4.86 | 2,789.31 | 170.71 | 13,167.83 | 71.49 | **30.17** |
| Qwen-2 | 151,665 | Go | 2.38 | 2,635.00 | 132.27 | 14,756.47 | X | **21.32** |
| | | Java | 2.39 | 2,734.27 | 260.80 | 15,235.25 | 50.32 | **25.12** |
| | | Python | 4.81 | 3,268.09 | 155.32 | 15,146.38 | 70.87 | **32.30** |

While of course we cannot guarantee that GREATGRAMMA is free of bugs, the simplicity of our approach makes its implementation easier (just 900 lines of code).

Furthermore, GREATGRAMMA's offline preprocessing times have allowed us to heavily test our implementation without incurring in multi-hour testing times.

**Limitations.** The current implementation of GREAT-GRAMMA works under the lexing assumptions described in Sec. 3.1: maximal munch principle and 1-lookahead lexing. It therefore does not support languages for which lexing requires more than 1-lookahead or instances where the same input sequence must be lexed differently depending on the parsing context.

For example, in Java, the end of the nested generic `List<List<T>>` is erroneously tokenized by a lexer operating under maximal munch as `>>`—i.e., the bitwise right-shift operator. However, in this context the parser instead expects two consecutive `>` terminals denoting the closure of a generic type. (The Java grammar we used in our evaluation, sourced from the SYNCODE benchmark, does not support generic types.)

To resolve this type of lexing ambiguity, the lexer must consider the current state of the parser, a task that for arbitrary regular expressions requires the lexer to perform unbounded lookahead. It is possible to modify our approach to handle practically occuring ambiguities by allowing the lexing transducer to nondeterministically generate both possible tokenizations (e.g., a single `>>` terminal or two separate `>` terminals), specifically for such cases that do not require arbitrary lookahead, and enabling the parser to select the valid interpretation based on grammatical constraints.

For a lexer requires $k > 1$ lookahead, a $k$-lookahead lexing transducer can be implemented by encoding the state and its $k$-lookahead as a single combined state, similar to an LR($k$) parser (Knuth, 1965). However, this approach can significantly increase the size of the resulting transducer.

While our implementation uses a deterministic PDA as its basis, our algorithm can be modified to handle nondeterminism by tracking multiple active states concurrently. This extension is applicable to nondeterministic PDAs without $\epsilon$-transition loops that push stack symbols. Handling multiple states involves calculating the possible next tokens for each state individually and then taking the union of these allowed tokens.

# 6. Related Work

There are many different implementations of grammar constrained decoding (Geng et al., 2023; Willard & Louf, 2023; Lundberg et al.; Beurer-Kellner et al., 2024; Ugare et al., 2024; Dong et al., 2024; Gerganov, 2024), but we focus our comparison on the ones that can handle realistic grammars (e.g., those of modern programming languages).

SYNCODE (Ugare et al., 2024) is the only that can provide low online overhead for the large grammars in our evaluation. SYNCODE speculatively unrolls future terminal sequences up to a fixed bound (i.e., every terminal sequence in $\Gamma^k$) and precomputes a table similar to our inverse token spanner table but *for all* sequences. Our evaluation shows that our table can be computed significantly faster since it only contains the set of realizable terminal sequences.

DOMINO (Beurer-Kellner et al., 2024) precomputes a tree of terminal sequences similar to our lexing transducer. However, at decoding time, DOMINO must traverse the entire tree because the set of realizable terminal sequences varies *for each* lexer state. In contrast, our approach computes a single *global* set of realizable terminal sequences $Re_{\mathcal{A} \circ \mathcal{V}}$, which we then use to preprocess the parser. DOMINO's implementation is not publicly available, but their paper reports 20s for offline preprocessing and 22% online overhead for

an extremely simplified C grammar with approximately 70 rules when using the smaller Llama tokenizer. For the same tokenizer, GREATGRAMMA can preprocess *much larger grammars* (353-520 rules) with similar times (25-35s).

XGRAMMAR (Dong et al., 2024) reduces runtime checks by preprocessing context-independent tokens for character-based nondeterministic PDAs, but mentions the misalignment problem in their related work. By preprocessing the set of realizable terminal sequences and the inverse token spanner table, we efficiently integrate their context-independent token caching approach with a parser that uses a separate lexer. Furthermore, our overapproximation via FSA can identify more always-rejected sequences compared to their expanded suffix analysis, as our method can also follow rule-reduction edges by treating them as $\epsilon$-transitions.

Existing research on the effectiveness of constrained decoding in structured generation shows mixed results. Some studies (Geng et al., 2023; Beurer-Kellner et al., 2024) demonstrate that GCD can improve downstream task performance. However, other recent work (Tam et al., 2024; Park et al., 2024) highlights that GCD can also negatively impact the quality of generated outputs. Our approach is orthogonal to these methods and can be integrated with techniques proposed by Park et al. (2024) or Banerjee et al. (2025) to mitigate such negative effects.

**Comparison with LLGUIDANCE (Moskal et al., 2025)**
After we completed this paper, Moskal et al. (2025) released LLGUIDANCE, a fast GCD tool written in Rust that incorporates a memory-optimized tokenizer trie, a derivative-based lazy lexer, and a highly optimized Earley parser. LLGUIDANCE shares some similarities with our compositional approach, but it relies on derivative-based parsing instead of automata and transducers constructions.

The evaluation provided on LLGUIDANCE's website reports impressive performance (in the order of micro to milliseconds for both offline pre-processing and online per-token overhead) on benchmarks like JSONSchemaBench (Geng et al., 2025). However, we could not compare this tool to our approach because LLGUIDANCE (v.0.7.27) reports the "ParserTooComplex" error when applied to the grammars used in our experiments. However, we believe that some of the optimizations employed by LLGUIDANCE can be also beneficial for GREATGRAMMA.

## 7. Conclusion

We present GREATGRAMMA, a tool for grammar-constrained decoding that is both flexible (low offline overhead) and efficient (low online overhead). GREATGRAMMA precomputes a succinct and easy-to-generate data structure that only captures terminal sequences that are *realizable* by

LLM tokens in a given lexer state. This data structure also speeds up online masking compared to similar approaches by further reducing the number of inputs the parser has to check during decoding.

GREATGRAMMA is built with simple primitives that are already supported by existing parsing libraries and lends itself to an easy implementation consisting of just 800 lines of Python. This simple implementation is easier to test (we have identified that other tools often crash or produce incorrect outputs) and opens the door to many future possible research directions—e.g., finding ways to leverage the PDA stack to perform more aggressive precomputation or implementing decoding directly on GPUs.

## Acknowledgments

Supported, in part, by a Microsoft Faculty Fellowship; a UCSD JSOE Scholarship; and NSF under grants CCF-2422214, CCF-2402833, and CCF-2211968. Any opinions, findings, and conclusions or recommendations expressed in this publication are those of the authors, and do not necessarily reflect the views of the sponsoring entities. Loris D'Antoni holds concurrent appointments as a Professor at the University of California San Diego and as an Amazon Scholar. This paper describes work performed at the University of California San Diego and is not associated with Amazon.

## Impact Statement

This paper presents work whose goal is to advance the field of Machine Learning. There are many potential societal consequences of our work, none which we feel must be specifically highlighted here.

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

## A. Algorithms

In this section we formalize several algorithms presented in the paper. Alg. 1 describes the abstract algorithm for general constrained decoding. 2 describes how to build the lexing transducer from a lexing automaton given as a FSA. Alg. 3 describes the construction of the detokenizing transducer, which converts sequences of tokens to sequences of the characters they contain. Alg. 5 describes the parser preprocessing and how the always-accepted token table and context-dependent sequence table are built.

---

**Algorithm 1** CONSTRAINEDDECODING

---

**Input:** Model $M$, Checker $C$, Tokenized prompt $x$
$\mathcal{V} := M.\text{vocabulary}$
**repeat**
   $m := C(x; \mathcal{V})$
   $logits := M(x)$
   $t_{next} := \text{sample}(\text{apply\_mask}(m, logits))$
   $x := x.\text{append}(t_{next})$
**until** $t_{next} \neq \text{EOS}$
**return** $x$

---

---

**Algorithm 2** BUILDLEXINGFST

---

**Input:** FSA $\mathcal{A} = (\Sigma, Q, q_0, \delta, F)$, Output alphabet $\Gamma$
**Output:** FST $\mathcal{T}_\mathcal{A} = (\Sigma, \Gamma, Q, q_0, \delta_{\text{FST}}, F_{\text{FST}})$
$\delta_{\text{FST}} := \{q \xrightarrow{c:\epsilon} q' \mid q \xrightarrow{c} q' \in \delta\}$, $F_{\text{FST}} := \{q_0\}$
**for** state $q$ that recognizes language token $T \in \Gamma$ **do**
   **for** $(c, q')$ s.t. $\exists q''. q \xrightarrow{c} q'' \notin \delta$ and $q_0 \xrightarrow{c} q' \in \delta$ **do**
      $\delta_{\text{FST}} := \delta_{\text{FST}} \cup \{q \xrightarrow{c:T} q'\}$
   $\delta_{\text{FST}} := \delta_{\text{FST}} \cup \{q \xrightarrow{\text{EOS}:T\$} q_0\} \cup \{q_0 \xrightarrow{\text{EOS}:\$} q_0\}$
**return** $\mathcal{T}_\mathcal{A} = (\Sigma, \Gamma, Q, q_0, \delta_{\text{FST}}, F_{\text{FST}})$

---

---

**Algorithm 3** BUILDDETOKENIZINGFST

---

**Input:** Vocabulary $\mathcal{V} \subseteq \Sigma^+$
**Output:** FST $\mathcal{T}_\mathcal{V} = (\mathcal{V}, \Sigma, Q, q_0, \delta, F)$
$Q := \{q_\epsilon\}$, $F := \{q_\epsilon\}$, $q_0 := q_\epsilon$, $\delta := \emptyset$
**for** $c_1 \ldots c_k \in \mathcal{V}$ **do**
   $q_{prev} := q_\epsilon$
   **for** $i = 1$ to $k - 1$ **do**
      $Q := Q \cup \{q_{c_1 \ldots c_i}\}$
      $\delta := \delta \cup \{q_{prev} \xrightarrow{\epsilon:c_i} q_{c_1 \ldots c_i}\}$
      $q_{prev} := q_{c_1 \ldots c_i}$
   $\delta := \delta \cup \{q_{prev} \xrightarrow{c_1 \ldots c_k : c_k} q_\epsilon\}$
**return** $\mathcal{T}_\mathcal{V} = (\mathcal{V}, \Sigma, Q, q_0, \delta, F)$

---

## B. Formal Definitions

### B.1. Finite-State Automata

A *finite-state automaton* (FSA) is defined as a tuple $\mathcal{A} = (\Sigma, Q, q_0, \delta, F)$ where $\Sigma$ is the input alphabet, $Q$ is the set of states, $q_0 \in Q$ is the initial state, $\delta \subseteq Q \times (\Sigma \cup \{\epsilon\}) \times Q$ is the set of transitions, and $F \subseteq Q$ is the set of accepting states. Each transition $(q, c, q')$, also denoted by $q \xrightarrow{c} q'$, indicates that, from state $q$, upon reading the input symbol $c$, the automaton transitions to state $q'$.

---

**Algorithm 5** PREPROCESSPARSER

---

**Input:** PDA $\mathcal{P}$, Realizable sequences $Re_{\mathcal{A} \circ \mathcal{V}}$,
FSA $\mathcal{A}$, Inverse token spannar table $T_{\text{inv}}$
**Output:** Always-accepted token table $A$
Context-dependent sequence table $D$
$\mathcal{A}_{\mathcal{P}} := \text{REMOVESTACKOPERATIONS}(\mathcal{P})$
**for** $q_{\mathcal{P}} \in Q_{\mathcal{P}}$ **do**
  $\bar{A}(q_{\mathcal{P}}) := \{\alpha \in Re_{\mathcal{A} \circ \mathcal{V}} \mid q \text{ with stack } \epsilon \text{ accepts } \alpha\}$
  $\bar{R}(q_{\mathcal{P}}) := \{\alpha \in Re_{\mathcal{A} \circ \mathcal{V}} \setminus A(q) \mid \mathcal{A}_{\mathcal{P}} \text{ rejects } \alpha \text{ in } q\}$
  $\bar{D}(q_{\mathcal{P}}) := Re_{\mathcal{A} \circ \mathcal{V}} \setminus A(q) \setminus R(q)$
  **for** $q_{\mathcal{A}} \in Q_{\mathcal{A}}$ **do**
    $A(q_{\mathcal{A}}, q_{\mathcal{P}}) := \bigcup_{\alpha \in \bar{A}(q)} T_{\text{inv}}(q_{\mathcal{A}}, \alpha)$
    $D(q_{\mathcal{A}}, q_{\mathcal{P}}) := \{\alpha \in \bar{D}(q) \mid T_{\text{inv}}(q_{\mathcal{A}}, \alpha) \neq \emptyset\}$
**return** $A, D$

---

Given a transition relation $\delta$ of automaton $\mathcal{A}$, the extended transition $\delta^* \subseteq Q \times \Sigma \times Q$ is the smallest relation defined by
(*i*) $(q, \epsilon, q) \in \delta^*$ and (*ii*) $(q, wc, q') \in \delta^*$ whenever $(q, w, q'') \in \delta^*$ and $(q'', c, q) \in \delta$ for some $q'' \in Q$. We also denote
$(q, w, q') \in \delta^*$ by $q \xrightarrow{w} q'$. A string $w \in \Sigma^*$ is accepted by automaton $\mathcal{A}$ when there exists $q' \in F$ such that $q \xrightarrow{w}{}^* q' \in \delta^*$.

However, in this paper, we use the term *accepted* in a broader sense than its standard definition. Specifically, we say that a string $w$ is accepted in state $q$ if there exists $q'$ such that $q \xrightarrow{w} q' \in \delta^*$, meaningthat a valid transition for $w$ exists from $q$.

### B.2. Context-Free Grammar

A *context-free grammar* (CFG) $\mathcal{G}$ is a tuple $(\Gamma, \mathcal{N}, S, \mathcal{R})$ where $\Gamma$ is a finite set of terminal symbols (e.g. constants, variable names, and keywords), $\mathcal{N}$ is a finite set of non-terminal symbols, $S \in \mathcal{N}$ is a start nonterminal, $\mathcal{R}$ is a set of production rules $A \to \alpha_1, \ldots, \alpha_n$ where $A \in \mathcal{N}$ and $\alpha_i \in \mathcal{N} \cup \Gamma$.

Formally, a grammar $\mathcal{G}$ defines a *single-step derivation* relation on sequences of symbols $\alpha, \beta, \gamma \in (\mathcal{N} \cup \Gamma)^*$: $\alpha A \gamma \Rightarrow \alpha \beta \gamma$ if $A \to \beta \in \mathcal{R}$. The reflexive transitive closure of this relation is called *derivation* and written $\Rightarrow^*$. A sequence of terminals $w$ is a *sentence* if it is derivable from $S$; the set of all sentences is called the *language* of the grammar $\mathcal{G}$, that is, $\mathcal{L}(\mathcal{G}) = \{w \in \Gamma^* \mid S \Rightarrow^* w\}$.

In addition, we define the *prefix language* of $\mathcal{G}$ as the set of all prefixes of sentences in $\mathcal{L}(\mathcal{G})$, that is, $\mathcal{L}_{\text{prefix}}(\mathcal{G}) = \{w \in \Gamma^* \mid \exists v \in \Gamma^*. wv \in \mathcal{L}(\mathcal{G})\}$.

### B.3. Finite State Transducer

An FST is a tuple $\mathcal{T} = (\Sigma, \Gamma, Q, q_0, \delta, F)$ where all components are analogous to those of a FSA but each FST transition $q \xrightarrow{c:T_1 \ldots T_n} q'$ in $\delta$ denotes that when reading a character $c$ from state $q$, the FST moves to a new state $q'$ and also outputs the sequence $T_1 \ldots T_k$ of elements over the output alphabet $\Gamma$.

### B.4. Pushdown Automata

A *pushdown automaton* is a tuple $\mathcal{P} = (\Sigma, \Pi, Q, q_0, Z_0, \delta, F)$ where $\Sigma$, $Q$, $q_0$ and $F$ are as in their FSA definitions, $\Pi$ is the stack alphabet, $Z_0 \in \Pi$ is the initial stack symbol, and $\delta \subseteq Q \times (\Sigma \cup \{\epsilon\}) \times \Pi^* \times Q \times \Pi^*$ is the set of transitions. Each transition $(q, c, \alpha, q', \beta)$ specifies that, in state $q$, upon reading the input symbol $c$ and matching the top stack symbols to $\alpha$, the PDA transitions to state $q'$ and replaces $\alpha$ with the sequence $\beta$ on the stack.

## C. Formalizations and Proofs

### C.1. Lexer

A *lexer specification* is given as a finite set of pairs, each consisting of an automaton $\mathcal{A}^i = (Q^i, \Sigma^i, q_0^i, \delta^i, F^i)$ and a terminal symbol $T^i \in \Gamma$. Given a lexer specification $\{(\mathcal{A}^i, T^i)\}_i$, a corresponding (1-lookahead) *partial lexer* is a partial function

Lex $: \Sigma^* \rightarrow \Gamma^* \times \Sigma^*$ in which defined as following:

- Lex$(\epsilon) = (\epsilon, \epsilon)$

- Given Lex$(w) = (T_1 \ldots T_k, w_r)$, the value of Lex$(wc)$ is

  (1) $(T_1 \ldots T_k T^j \$, \epsilon)$ if $c = $ EOS and $w_r \in \mathcal{L}(\mathcal{A}^j)$ for some $j$
  (2) $(T_1 \ldots T_k, w_r c)$ if $c \neq $ EOS and $w_r c \in \mathcal{L}_{\text{prefix}}(\mathcal{A}^i)$ for some $i$
  (3) $(T_1 \ldots T_k T^j, c)$ if $c \neq $ EOS and $w_r \in \mathcal{L}(\mathcal{A}^j)$ but $w_r c \notin \mathcal{L}_{\text{prefix}}(\mathcal{A}^i)$ for all $i$
  (4) $\perp$ otherwise

The lexing automaton $\mathcal{A}$ is defined as the product automaton of automata $\mathcal{A}^i$ specified in the lexer definition $\{(\mathcal{A}^i, T^i)\}$.

**Lemma C.1.** *Let $\mathcal{T}_\mathcal{A} = (Q, \Sigma, \Gamma, q_0, \delta, F)$ be a lexing transducer for the lexer specification $\{(\mathcal{A}^i, T^i)\}_i$. Assume $\epsilon \notin \mathcal{L}(\mathcal{A}^i)$ for all $i$ and each $\mathcal{A}^i$ does not have any dead state. Then, for any string $w \in \Sigma^*$, Lex$(w) = (T_1 \ldots T_k, w_r)$ if and only if $q_0 \xrightarrow{w:T_1 \ldots T_k}^* q' \in \delta^*$ and $q_0 \xrightarrow{w_r:\epsilon}^* q' \in \delta^*$ for some $q' \in Q$.*

*Proof.* The proof proceeds by induction. For the base case, let $w = \epsilon$. Then Lex$(\epsilon) = (\epsilon, \epsilon)$ and $q_0 \xrightarrow{\epsilon:\epsilon}^* q_0 \in \delta^*$. This is the only possible transition for empty input, as $\epsilon \notin \mathcal{L}(\mathcal{A}^i)$ for each $i$.

Otherwise, we consider the remaining cases in the definition of Lex. Cases (1)-(3) are by construction. Assume Lex$(wr) = (T_1 \ldots T_k, w_r)$, then by the induction hypothesis, $q_0 \xrightarrow{w:T_1 \ldots T_k}^* q' \in \delta^*$ and $q_0 \xrightarrow{w_r:\epsilon}^* q' \in \delta^*$ for some $q' \in Q$. The final case applies when Lex$(wc) = \perp$, indicating that no valid tokenization of $wc$ can be produced. By assumption, our component automata are constructed without dead states. This ensures that there doesn't exists $q' \in Q$ such that $q_0 \xrightarrow{w_r:T_1 \ldots T_k}^* q' \in \delta^*$ when $w_r \notin \mathcal{L}_{\text{prefix}}(\mathcal{A}^i)$ for all $i$. Conversely, if there doesn't exist $q' \in Q$ such that $q_0 \xrightarrow{w:T_1 \ldots T_k}^* q' \in \delta^*$, then by construction, cases (1)-(3) cannot hold so Lex$(wc)$ must be $\perp$.

$\square$

The correctness of the detokenizing transducer is proven in the Appendix of (Koo et al., 2024). The correctness of the token-level lexing transducer follows from the correctness of transducer composition.

**Theorem C.2.** *Let $\mathcal{T}_{\mathcal{A} \circ \mathcal{V}} = (Q, \mathcal{V}, \Gamma, q_0, \delta, F)$ be a token-level lexing transducer for the lexer specification $\{(\mathcal{A}^i, T^i)\}_i$ and vocabulary $\mathcal{V} \subseteq \Sigma^+$. Then $q_0 \xrightarrow{w:T_1 \ldots T_k}^* q' \in \delta^*$ if and only if Lex$(w) = (T_1 \ldots T_k, w_r)$ for some $w_r \in \Sigma^*$ and $q' \in Q$ such that $q_0 \xrightarrow{w_r:\epsilon}^* q'$.*

### C.2. Inverse Token Spanner Table

By construction of inverse token spanner table, the following proposition holds.

**Proposition C.3.** *Given a token sequence $w \in \mathcal{V}^*$ such that $q_0 \xrightarrow{w:T_1 \ldots T_k}^* q'$, any $v \in \mathcal{V}$ and $T \in \Gamma$, the token $v$ is in the inverse token spanner table entry $T_{inv}(q', T_{k+1} \ldots T_m T)$ if and only if $q_0 \xrightarrow{wv:T_1 \ldots T_k T_{k+1} \ldots T_m}^* q''$ and $T \in Prod(q'')$.*

### C.3. Parser

Given a context-free grammar $\mathcal{G} = (\Gamma, \mathcal{N}, S, \mathcal{R})$ with a separate lexer Lex $: \Sigma^* \rightarrow \Gamma^* \times \Sigma^*$, we say that a token sequence $w \in \mathcal{V}^*$ is a *sentence* if there exists a derivable sequence of terminals $T_1 \ldots T_k$ such that Lex$(w) = (T_1 \ldots T_k \$, \epsilon)$. Formally,

$$\mathcal{L}^{\text{Lex}}(\mathcal{G}) = \{w \in \mathcal{V}^* \mid \exists T_1 \ldots T_k. \text{Lex}(w) = (T_1 \ldots T_k \$, \epsilon) \wedge T_1 \ldots T_k \$ \in \mathcal{L}(\mathcal{G})\}$$

We also define the *prefix language* of $\mathcal{G}$ with separate lexer Lex as the set of all prefixes of sentences in $\mathcal{L}^{\text{Lex}}(\mathcal{G})$, that is, $\mathcal{L}^{\text{Lex}}_{\text{prefix}}(\mathcal{G}) = \{w \in \Sigma^* \mid \exists v \in \Sigma^*. wv \in \mathcal{L}^{\text{Lex}}(\mathcal{G})\}$.

**Lemma C.4.** *Consider a string $w \in \mathcal{V}^*$ with an extended transition $q \xrightarrow{w:T_1 \ldots T_k}^* q'$ through the lexing transducer, where $T_1 \ldots T_k$ is incomplete prefix of $\mathcal{L}(\mathcal{G})$ (i.e., $T_1 \ldots T_k \in \mathcal{L}_{\text{prefix}}(\mathcal{G}) \setminus \mathcal{L}(\mathcal{G})$). If $w \in \mathcal{L}^{\text{Lex}}_{\text{prefix}}(\mathcal{G})$ then there must exist some terminal $T \in Prod(q')$ such that $T_1 \ldots T_k T \in \mathcal{L}_{\text{prefix}}(\mathcal{G})$.*

*Proof.* Suppose $w \in \mathcal{L}_{\text{prefix}}^{\text{Lex}}(\mathcal{G})$. By definition of the prefix language, there exists a non-empty suffix $v \in \mathcal{V}^+$ such that $wv \in \mathcal{L}^{\text{Lex}}(\mathcal{G})$. Consequently, $\text{Lex}(wv)$ must yield a complete token sequence ending in a EOS token, say $(T_1 \ldots T_k T_{k+1} \ldots T_m \$, \epsilon)$, where $T_1 \ldots T_k T_{k+1} \ldots T_m \$ \in \mathcal{L}(\mathcal{G})$.

Given our assumption that $T_1 \ldots T_k \notin \mathcal{L}(\mathcal{G})$, it follows that $T_{k+1} \ldots T_m$ cannot be empty. Specifically, this means $T_{k+1}$ must exist and be producible from the state $q'$, i.e., $q' \in Prod(q')$.

Therefore, by combining $T_1 \ldots T_k$ with the next token $T_{k+1}$, we form a valid prefix of a complete token sequence $T_1 \ldots T_k T_{k+1} \in \mathcal{L}_{\text{prefix}}(\mathcal{G})$. $\square$

**Theorem C.5** (Completeness). *Let* $\mathcal{P} = (\Gamma, \Pi, Q^{\mathcal{P}}, q_0^{\mathcal{P}}, Z_0, \delta^{\mathcal{P}}, F^{\mathcal{P}})$ *be a pushdown automaton such that* $\mathcal{L}(\mathcal{P}) = \mathcal{L}(\mathcal{G})$. *Given a string* $w \in \mathcal{L}_{\text{prefix}}^{\text{Lex}}(\mathcal{G})$ *and a token* $v \in \mathcal{V}$, *the token* $v$ *is not masked by Alg. 6 (i.e.,* $v \in \mathcal{V}_{allowed}$*) if the concatenation* $wv$ *belongs to* $\mathcal{L}_{\text{prefix}}^{\text{Lex}}(\mathcal{G})$.

*Proof.* By Theorem C.2, the lexing transducer emitted $T_1 \ldots T_k$ and reached a state $q'$ such that $q_0 \xrightarrow{w_r:\epsilon}^* q'$. If $wv \in \mathcal{L}_{\text{prefix}}^{\text{Lex}}(\mathcal{G})$, then $q_0 \xrightarrow{wv:T_1 \ldots T_k T_{k+1} \ldots T_m} q''$ for some state $q''$, which implies $q' \xrightarrow{T_{k+1} \ldots T_m} q''$. By Lemma C.4, there exists $T \in Prod(q'')$ such that $T_1 \ldots T_k T_{k+1} \ldots T_m T \in \mathcal{L}_{\text{prefix}}(\mathcal{G})$. Therefore, $v \in T_{\text{inv}}(q', T_{k+1} \ldots T_m T)$, and hence $v \in \mathcal{V}_{allowed}$. $\square$

**Lemma C.6.** *If* $T \in Prod(q)$, *then there exist* $q' \in Q$ *and* $v \in \mathcal{V}^*$ *such that* $q \xrightarrow{v:T}^* q' \in \delta^*$.

*Proof.* Since $T \in Prod(q')$, there exist $q'' \in Q$ and $v \in \mathcal{V}^*$ such that $q' \xrightarrow{v:T\alpha}^* q'' \in \delta^*$. Note that $\Sigma \subseteq \mathcal{V}$. A key property of the lexing transducer is that it produces at most one terminal for each input character. Consequently, there must exist a prefix $v_p$ of $v$ and a state $q''' \in Q$ such that $q' \xrightarrow{v_p:T}^* q''' \in \delta^*$. $\square$

**Lemma C.7.** *Assume that if there exists some* $v \in \mathcal{V}^*$ *such that* $q_0 \xrightarrow{w_r v:T} q$, *then for any terminal sequence* $\alpha \in \Gamma^*$, *there also exists some* $v' \in \mathcal{V}^*$ *such that* $q_0 \xrightarrow{w_r v':T\alpha} q'$. *Consider a string* $w \in \mathcal{V}^*$ *with an extended transition* $q \xrightarrow{w:T_1 \ldots T_k}^* q'$ *through the lexing transducer, where* $T_1 \ldots T_k$ *is incomplete prefix of* $\mathcal{L}(\mathcal{G})$ *(i.e.,* $T_1 \ldots T_k \in \mathcal{L}_{\text{prefix}}(\mathcal{G}) \setminus \mathcal{L}(\mathcal{G})$*). If there exists* $T \in Prod(q')$ *such that* $T_1 \ldots T_k T \in \mathcal{L}_{\text{prefix}}(\mathcal{G})$ *then* $w \in \mathcal{L}_{\text{prefix}}^{\text{Lex}}(\mathcal{G})$.

*Proof.* Suppose there exists $T \in Prod(q')$ such that $T_1 \ldots T_k T \in \mathcal{L}_{\text{prefix}}(\mathcal{G})$. By the definition of prefix language, there exists $\alpha \in \Gamma^*$ such that $T_1 \ldots T_k T\alpha \in \mathcal{L}(\mathcal{G})$. Furthermore, by Lemma C.6, for such a $T$, there exist some state $q' \in Q$ and a string $v \in \mathcal{V}^*$ such that $q' \xrightarrow{v:T}^* q'' \in \delta^*$. Now, by the assumption, there also exists a string $v' \in \mathcal{V}^*$ such that for some state $q''' \in Q$, $q' \xrightarrow{wv':T_1 \ldots T_k T\alpha}^* q''' \in \delta^*$. Here, since $\alpha$ must end with \$, it implies that $q'''$ must be the initial state $q_0$ by construction. Thus, we have $\text{Lex}(wv') = (T_1 \ldots T_k T\alpha, \epsilon)$. Since $T_1 \ldots T_k T\alpha \in \mathcal{L}(\mathcal{G})$, this final result implies that $w \in \mathcal{L}_{\text{prefix}}^{\text{Lex}}(\mathcal{G})$. $\square$

**Theorem C.8** (Soundness). *Let* $\mathcal{P} = (\Gamma, \Pi, Q^{\mathcal{P}}, q_0^{\mathcal{P}}, Z_0, \delta^{\mathcal{P}}, F^{\mathcal{P}})$ *be a pushdown automaton such that* $\mathcal{L}(\mathcal{P}) = \mathcal{L}(\mathcal{G})$. *Given a string* $w \in \mathcal{L}_{\text{prefix}}^{\text{Lex}}(\mathcal{G})$ *and a token* $v \in \mathcal{V}$, *the token* $v$ *is not masked by Alg. 6 (i.e.,* $v \in \mathcal{V}_{allowed}$*) only if the concatenation* $wv$ *belongs to* $\mathcal{L}_{\text{prefix}}^{\text{Lex}}(\mathcal{G})$.

*Proof.* By Theorem C.2, the lexing transducer emitted $T_1 \ldots T_k$ and reached a state $q'$ such that $q_0 \xrightarrow{w_r:\epsilon}^* q'$. If $v \in \mathcal{V}_{allowed}$, then there must exist some $T \in \Gamma$ such that $v \in T_{\text{inv}}(q', T_{k+1} \ldots T_m T)$. By the construction of inverse token spanner table, this implies that the concatenation $T_1 \ldots T_k T_{k+1} \ldots T_m T$ is in the prefix language $\mathcal{L}_{\text{prefix}}(\mathcal{G})$. Furthermore, this means that there exist some $q'' \in Q$ such that $q' \xrightarrow{T_{k+1} \ldots T_m} q''$ and $T \in Prod(q'')$. Therefore, by Lemma C.7, the concatenation $wv$ belongs to $\mathcal{L}_{\text{prefix}}^{\text{Lex}}(\mathcal{G})$.

$\square$

