# OpenReview forum: "Flexible and Efficient Grammar-Constrained Decoding"
_ICML.cc/2025/Conference — ICML 2025 poster_

### Official Review · Reviewer_EMYM · 2025-02-27

**Overall Recommendation:** 3

**Summary:**

- The paper presents a new algorithm for grammar-constrained decoding (GCD) that significantly improves the efficiency of both offline preprocessing and online token masking. The key innovation is a combined analysis of the LLM token vocabulary and set of CFG terminals, which precomputes a lexer-state-dependent mapping between sequences of CFG tokens and individual LLM tokens.
- The proposed method, implemented in a tool called GREATGRAMMA, achieves an average 17.71x speedup in offline preprocessing compared to related approaches while maintaining state-of-the-art online masking efficiency (5-32ms per token).
- The approach is flexible, handling diverse grammars without prohibitive offline preprocessing costs, and efficient, maintaining low online token-masking overhead.

**Claims And Evidence:**

- The claims made in the paper are generally supported by clear and convincing evidence. The authors provide detailed algorithm descriptions, theoretical analysis, and experimental results.
- They demonstrate the effectiveness of their approach through comparisons with existing GCD tools (OUTLINES, SYNCODE, and XGRAMMAR) on several benchmarks.
- For instance, Table 1 shows that GREATGRAMMA's offline preprocessing is significantly faster than SYNCODE and its online masking is faster than both SYNCODE and OUTLINES.

**Essential References Not Discussed:**

- The paper references several related works on grammar-constrained decoding (GCD), including SYNCODE, DOMINO, and XGRAMMAR. However, it does not discuss other relevant works such as Grammar-Aligned Decoding (Park et al., 2024) or Synchromesh (Poesia et al., 2022), which also address the problem of aligning LLM outputs with grammatical constraints. These works provide additional context and comparisons that could enhance the understanding of GREATGRAMMA's contributions.
   - The paper also does not mention the work on constrained decoding using finite-state transducers (FSTs) by Koo et al. (2024), which is directly relevant to the preprocessing and token masking techniques used in GREATGRAMMA.

**Experimental Designs Or Analyses:**

- The experimental designs are sound and valid. The authors conducted experiments using three different tokenizers and several programming language grammars.
- They measured both offline preprocessing time and online per-token overhead, which are critical metrics for evaluating GCD tools.
- The results are presented in a clear and organized manner, with tables and figures that effectively communicate the findings.

**Methods And Evaluation Criteria:**

- The proposed methods make sense for the problem at hand. The approach addresses the key challenge of token misalignment between LLMs and CFGs by preprocessing the lexer and parser in tandem.
- The evaluation criteria, including offline preprocessing time and online per-token overhead, are appropriate for assessing the performance of GCD tools.

**Other Comments Or Suggestions:**

No obvious typographical errors were observed in the provided text.

**Other Strengths And Weaknesses:**

- **Strengths:**
    - **Originality:** The approach proposed in the paper, particularly the combined analysis of the LLM token vocabulary and CFG terminals, as well as the introduction of the token spanner table, is novel. This innovation addresses a significant challenge in the field of grammar-constrained decoding.
    - **Significance:** The results are important for improving the efficiency of constrained decoding in large language models, which has practical implications for applications like code generation and structured data formatting.
    - **Clarity:** The paper is well-written and well-organized. The authors provide a clear explanation of the problem, their approach, algorithms, and experimental results. The use of figures to illustrate the lexing transducer, detokenizing transducer, and token spanner table helps in understanding the complex concepts.
- **Weaknesses:**
    - **Limitations:** As mentioned in the paper, the current implementation has some limitations. For example, it works under the maximal munch principle and 1-lookahead lexing, which may not support all language constructs. Additionally, the approach may not handle cases where the same input sequence must be lexed differently depending on the parsing context.
    - **Theoretical Depth:** While the paper provides an algorithmic and conceptual description of the proposed method, some theoretical aspects could be explored further. For example, a more in-depth analysis of the properties of the realizable terminal sequences and their impact on the overall decoding process could be beneficial.
    - **Application Scope:** The paper primarily focuses on grammar-constrained decoding for structured outputs like code snippets. However, the approach could be extended to other domains, such as natural language generation with specific syntax requirements, which is not discussed in detail.

**Questions For Authors:**

1. Regarding the limitations in handling lexing ambiguities, could you elaborate on how you plan to extend the approach to support more complex language constructs that require more than 1 - lookahead or context - dependent lexing? How would a successful extension impact the performance and applicability of GREATGRAMMA?
    - If the authors have a clear plan and potential solutions for handling these cases, it would strengthen the paper's future work section and indicate a better understanding of the problem domain.
2. In the experimental evaluation, you mentioned that XGRAMMAR encountered errors during preprocessing or decoding for all the grammars considered. Could you provide more details about these errors and how they might affect the comparison with GREATGRAMMA?
    - Understanding the specific issues with XGRAMMAR would help in assessing the fairness of the comparison and determining whether GREATGRAMMA's advantages are as significant as reported.
3. You mentioned that the current implementation works under certain lexing assumptions. How would you evaluate the potential benefits and challenges of relaxing these assumptions to handle a broader range of language grammars?
    - The authors' perspective on this could provide insight into the scalability and generalizability of their approach.

**Relation To Broader Scientific Literature:**

- The key contributions of the paper are related to several prior works in the field of grammar-constrained decoding. The authors build upon existing approaches such as SYNCODE and DOMINO.
- Compared to SYNCODE, GREATGRAMMA offers faster offline preprocessing and similar or better online masking efficiency.
- The approach also integrates ideas from XGRAMMAR, such as preprocessing context-independent tokens, but with improved efficiency and scalability.

**Theoretical Claims:**

- The paper does not present detailed proofs for their theoretical claims, but the algorithms and concepts are well-founded.
- The authors provide references to existing literature for some of the theoretical foundations, such as finite-state automata and pushdown automata.
- The correctness of the algorithms is mostly supported by the experimental results.

---

> ### Author Rebuttal · Authors · 2025-04-01
>
> We thank the reviewer for their insightful feedback. Below, we provide a detailed response to each comment and question.
>
> > The paper does not present detailed proofs for their theoretical claims
> > a more in-depth analysis of the properties of the realizable terminal sequences and their impact on the overall decoding process could be beneficial
>
> We will provide a more rigorous formalization, correctness proofs, and time/space analysis in the appendix of the revised paper. For proof sketches regarding correctness, please refer to our response addressing a similar point raised by Reviewer apti and Reviewer nQQ5.
>
> >could you elaborate on how you plan to extend the approach to support more complex language constructs that require more than 1 - lookahead or context-dependent lexing?
> > You mentioned that the current implementation works under certain lexing assumptions. How would you evaluate the potential benefits and challenges of relaxing these assumptions to handle a broader range of language grammars?
>
> To be clear, maximal munch and 1-lookahead are specifications about how a lexer is expected to behave, not limitations of our system. Lexers for real-world languages usually have these properties (including the ones we evaluated on: Python, Java, and Go). That being said, our lexing transducer can also be adapted to accommodate alternative lexer specifications. For instance, the lexing transducer can nondeterministically produce all possible tokenizations, enabling contextual lexing. However, this nondeterminism may increase the size of $Re_{A \circ V}$ and the inverse token spanner table.
>
> If the lexer requires more than one-token lookahead, a k-token lookahead can be implemented by encoding each pair (state, lookahead) as a single combined state in the lexing transducer, similar to an LR(k) parser [NewRef3]. However, this approach can significantly increase the size of the lexing transducer.
>
> We will include this discussion in the revised version of the paper.
>
> [NewRef3] Knuth, Donald E. "On the translation of languages from left to right." Information and control 8.6 (1965): 607-639.
>
> > In the experimental evaluation, you mentioned that XGRAMMAR encountered errors during preprocessing or decoding for all the grammars considered. Could you provide more details about these errors and how they might affect the comparison with GREATGRAMMA?
>
> XGrammar encountered a segmentation fault when decoding with the Go grammar, aborted without displaying a detailed error message after compiling the Java grammar for several minutes, and entered an infinite loop during the initial token generation step for the Python grammar. While we have not investigated the exact cause of these errors in XGrammar, we conjecture that they arise because XGrammar operates on byte-level pushdown automaton, which are significantly more complex than terminal-level pushdown automaton.
> Similar problems in XGrammar have also been observed with other grammars: https://github.com/mlc-ai/xgrammar/issues/127. According to the github issue, the runtime overhead increases significantly due to nondeterminism caused by the overlap between identifier and keyword terminals. Note that in GreatGramma the terminal-level PDA in this example is still deterministic: the nondeterminism is purely at the terminal-level, which is resolved by the separate lexer.
>
> > However, it does not discuss other relevant works such as Grammar-Aligned Decoding (Park et al., 2024) or Synchromesh (Poesia et al., 2022)
>
> The paper briefly mentions GAD in the context of evaluating the effectiveness of GCD (lines 330–333). While existing studies (Geng et al., 2023; Beurer-Kellner et al., 2024) demonstrate that GCD can improve downstream task performance, other recent work (Tam et al., 2024 [NewRef4]; Park et al., 2024) highlights that GCD can also negatively impact the quality of generated outputs. Our approach is orthogonal to these methods and can be integrated with techniques proposed by Park et al. (2024) or Banerjee et al. (2025) [NewRef5] to mitigate such negative effects. We will include this discussion on the effectiveness of GCD in Section 6.
>
> Syncromesh emphasizes checking semantic constraints beyond grammatical correctness, we will briefly discuss this extension as well.
>
> [NewRef4] Tam, Zhi Rui, et al. "Let Me Speak Freely? A Study On The Impact Of Format Restrictions On Large Language Model Performance." EMNLP 2024: Industry Track.
>
> [NewRef5] Banerjee, Debangshu, et al. "CRANE: Reasoning with constrained LLM generation." arXiv (2025).
>
> > The paper also does not mention the work on constrained decoding using FSTs by Koo et al. (2024)
>
> The paper briefly introduces the work of Koo et al. when presenting the detokenizing transducer in Section 3.1. However, we agree that this relevant work deserves further discussion. We extended their work to accommodate a structure consisting of a separate lexer and parser and will include this discussion in Section 6.

---

> > ### Comment · Reviewer_EMYM · 2025-04-07
> >
> > Thanks for the author's careful reply.
> >
> > According to the Github you provided, I found that there might be a problem in your reproduction process. The author did not modify the code and pointed out that the latest version might be needed (https://github.com/mlc-ai/xgrammar/issues/127#issuecomment-2777313764).
> >
> > In addition, I admit that I am not an expert in this field. What is the difference between online and offline use in actual scenarios? GREATGRAMMA is more than 10 times to more than 100 times slower than outlines in offline. Is this acceptable? I have used outlines in actual scenarios, and it is not very slow. (If time permits, can you provide a speed comparison in actual scenarios?)
> >
> > By the way, I have seen that many GCD-related papers have restricted generation of json or other formats.
> >
> > Finally, does this field only need one experiment to prove the effectiveness of the method, without any ablation and analysis experiments? If so, please ignore this comment.

---

> > > ### Author Response · Authors · 2025-04-08
> > >
> > > Thanks for pointing out that XGrammar has recently updated their code to *attempt* to fix this issue (we note that this update happened only a few days ago). While the update has removed the problem from the grammar used in the GitHub issue we linked (https://github.com/mlc-ai/xgrammar/issues/127) , this new version of XGrammar still causes errors or infinite loops for all the grammars used in our experiments. The reviewer can test this if they want by using the latest pip version (released on March 25, 2025) on the grammars we submitted as supplementary materials.
> > >
> > >
> > > Regarding offline versus online cost: offline costs are incurred once per grammar, while online costs apply to every inference.
> > >
> > >
> > > If an application uses a fixed grammar, a grammar can be precompiled by developers, making offline costs generally less critical than online costs for end users of the tool.
> > >
> > >
> > > However, offline costs become significant if the grammar must be changed frequently. For example, one might want to specialize a language grammar using specific program variables appearing in a concrete environment. In such scenarios, it is important to reduce offline preprocessing while retaining low online cost—a high online overhead can prevent large-scale batching. We do believe our grammars are practically useful as they can help reduce syntactic errors for real programming languages as shown in the SynCode paper. Even though the shortest example in our experiment is a 17-lines Python code (prime.py) containing around 100 tokens, running this example with Outlines already takes more than 1,000 seconds, which exceeds the offline cost of GREATGRAMMA.
> > >
> > >
> > > The reviewer is right that many recent works on constrained generation focus on JSON schemas. While today’s JSON schemas are often fairly simple, supporting larger, programming language–level grammars can provide enhanced reliability and flexibility for various applications, including tool integration mentioned earlier.
> > >
> > >
> > > Regarding evaluation, besides the quality of outputs generated via constrained decoding, which was addressed in our previous rebuttal, existing works—including SynCode and XGrammar—are evaluated based on offline cost and per-token online cost. As we use the same metrics as existing work, we believe that our evaluation setting is sufficient to demonstrate the efficiency of GREATGRAMMA. We can’t think of a potential ablation for our evaluation since we already compare GREATGRAMMA to existing tools (e.g., as discussed in our related work, SynCode is effectively GREATGRAMMA without the inverse spanner table).

---

### Official Review · Reviewer_nQQ5 · 2025-03-13

**Overall Recommendation:** 4

**Summary:**

The authors provide a more computationally efficient method, called GREATGRAMMA, for computing token masks for CFG-constrained decoding from LLMs. In particular, they address the misalignment between CFG terminals and the LLM token vocabulary, and tricks for speeding up online mask computation during decoding using relatively cheap preprocessing. Their method assumes that the constraint is expressed as a CFG whose terminal symbols are *chunks* of characters; these chunks are referred to as the "terminals." Their preprocessing step builds a FST that maps LLM token sequences to character strings and another FST that maps character strings to terminal sequences. They compose them to form a single FST that maps LLM token sequences to terminal sequences. They precompute a table that maps FST states and terminal sequences to valid LLM tokens. Their method is limited to grammars with certain properties (for one, they must be deterministic). They show through experiments that GREATGRAMMA's preprocessing is a little slower than the competing OUTLINES, but its online mask computation is much, much faster.

**Claims And Evidence:**

The empirical claims about runtime are well-supported by experiments.

On the other hand, the authors do not prove the correctness of their algorithms and do not provide asymptotic time/space complexity analysis, which I think is sorely needed.

**Essential References Not Discussed:**

Not to my knowledge.

Although light on detail, I believe this concurrent work uses a similar approach for token-character alignment by composing FSTs and CFGs: https://openreview.net/forum?id=xoXn62FzD0

**Experimental Designs Or Analyses:**

To an extent, yes, but I would like the authors to follow up with proofs of correctness.

**Methods And Evaluation Criteria:**

Yes.

**Other Comments Or Suggestions:**

1. The correctness of token alignment is a major issue. Can you formally express what the mathematically correct solution to the token alignment problem is?
2. 047 right: Interesting!
3. 087: Does this do things like strip whitespace and comments, as is done in a typical compiler? Or is the transformation bijective?
4. 511: This should check for t_next = EOS. Also, I don't think you want to append EOS to x.
5. 130: From this description, it's not clear what the meaning of a cell in the token spanner table is, much less how to construct it. It would be helpful to give a formal definition of what the value of each cell is supposed to be.
6. 204: Is $q_0$ a typo?
7. 520: What does it mean for state $q$ to recognize $T$? How do you determine what $T$ is?
8. 250: When you say $T$ can be generated along some path, do you mean a path ending in an accept state?
9. For some reason, the inputs and outputs of $\mathcal{T}\_{\mathcal{A}}$, $\mathcal{T}\_\mathcal{V}$, and $\mathcal{T}\_{\mathcal{A} \circ \mathcal{V}}$ are a bit unclear from reading the text, partly because $\mathcal{A} \circ \mathcal{V}$ is written in the reverse order of the inputs and outputs. It would be useful to spell this out in the text more explicitly. It took a while to figure out that $\mathcal{T}\_{\mathcal{A} \circ \mathcal{V}}$ maps sequences of LLM tokens to sequences of CFG terminals.
10. 249: There are some $\exists$ quantifiers missing.
11. 227: How is this loop implemented?
12. 251 right: Is $Re\_{\mathcal{A} \circ \mathcal{V}}$ a typo?
13. The wording of Proposition 3.4 is unclear. Do you mean if the PDA *starts* in state $q$ with stack $\gamma$?
14. The concepts of push and pop computations in PDAs from this paper may be useful to you: https://arxiv.org/abs/2210.06884
15. It's confusing that Fig 1(b) does not show an *inverse* table, but a table that maps states and *single tokens* to *sequences of terminals*. This causes a lot of confusion later on.
16. It wasn't clear until later in the paper that there are bounds on the size of the set of realizable tokens. This is important because otherwise it seems like the method requires exponential time/space.

**Other Strengths And Weaknesses:**

**Strengths**

1. The authors clearly state the limitations of their method.
2. The paper is generally well-written, although I have issues with aspects of the exposition.
3. The empirical results are very good.
4. Good benchmarks.

**Weaknesses**

1. As the authors state, the method is limited to *deterministic* CFLs that satisfy certain properties.
2. There are no proofs of correctness or time/space complexity analyses.
3. The exposition of the method is a bit unclear at times. In particular, their method hinges upon their concept of the set of realizable tokens, but the reason why this approach is correct and necessary is a particularly weak part of the exposition. See also Comments or Suggestions.

**Questions For Authors:**

1. Why is there a separation between characters and CFG terminals that is mediated by the lexer? It's always possible to implement the lexer directly within the CFG by modifying its rules and making its terminal symbols single characters. Is there some computational efficiency advantage to treating them separately?
2. 058: Shouldn't it also include a mask for EOS? How is EOS masking handled in your method?
3. What are the inputs and outputs of $\mathcal{T}\_{Re\_{\mathcal{A} \circ \mathcal{V}}}$?

**Relation To Broader Scientific Literature:**

They provide a faster, more practical method for CFG-constrained decoding than previous work, namely OUTLINES and SYNCODE. See their Section 6 for details.

**Theoretical Claims:**

This paper does not provide proofs for its theoretical claims, but I think it should prove the correctness of its algorithms and give a time/space complexity analysis.

---

> ### Author Rebuttal · Authors · 2025-04-01
>
> We thank the reviewer for their insightful feedback. In the revised paper, we will correct all typos, clarify the notation pointed out by the reviewer, and provide rigorous definitions and proofs in the appendix. Below, we provide a detailed response to each comment and question.
>
> > the method is limited to deterministic CFLs that satisfy certain properties
>
> Although we assumed a deterministic PDA for simplicity of implementation, determinism is not strictly necessary for our method. For more detail, please see our response to a similar question by reviewer q1Dz.
>
> > 227:How is this loop implemented?
>
> This line is precomputed by a reachability algorithm (e.g., Floyd-Warshall Algorithm [NewRef2]) in the lexing transducer.
>
> [NewRef2] Floyd, Robert W. "Algorithm 97: shortest path." Communications of the ACM 5.6 (1962)
>
> > no proofs of correctness or time/space complexity analyses
> > the reason why this approach is correct and necessary is a particularly weak part of the exposition
>
> Computing the set of realizable terminal sequences allows us to do exactly the amount of necessary work during parser preprocessing. We will improve the presentation to make this point clearer.
>
> For proof sketches regarding correctness, please refer to our response addressing a similar point raised by Reviewer apti. Regarding complexity, the most computationally expensive procedures are Algorithms 4 and 5.
>
> Algorithm 4 runs in time $O(|\delta||\Gamma|)$, in addition to the precomputation for line 227 which runs in $O(|Q_A|^3)$ time and $O(|Q_A|^2)$ space. Both $|Re_{A \circ V}|$ and $|T_{inv}|$ are at most $|Q_A||V||\Gamma|$, though empirically they tend to be much smaller since most combinations are infeasible.
> CFG parsing has time complexity $O(n^3)$ for general CFGs and $O(n)$ for deterministic CFGs. Consequently, computing $\bar{A}$ and $\bar{D}$ in Algorithm 5 takes $O(|Re_{A \circ V}| \cdot n^3)$ time for general CFGs and $O(|Re_{A \circ V}| \cdot n)$ time for deterministic CFGs, where $n$ denotes the length of the longest terminal sequence plus length of the current prefix.
> The computation of sets $A$ and $D$ in Algorithm 5 runs in $O(|Q_A||Q_P||V||\Gamma|)$ time and requires $O(|Q_A||Q_P||V|)$ space. In practice, time/space usage is often significantly smaller due to the sparsity in $T_{inv}$.
> In practice, we observe that $|Q_A|, |Q_P| < 1000$, $|\Gamma| < 100$ even for large grammars, including the Go, Java, and Python grammars used in our experiments.
>
> > Can you formally express what the mathematically correct solution to the token alignment problem is?
>
> Formally, given a context-free grammar $G$ and a lexer $Lex$, the token-aligned language $L^{Lex}(G)$ is defined as the set of $w \in V^\ast$ s.t. $\exists T_1 \dots T_k. Lex(w) = T_1 \dots T_k \wedge T_1 \dots T_k \in L(G)$. Correct token-aligned constrained decoding thus requires that any partial output is always a prefix of some sentence in $L^{Lex}(G)$.
>
> > it's not clear what the meaning of a cell in the token spanner table is
> > It's confusing that Fig 1(b) does not show an inverse table, but a table that maps states and single tokens to sequences of terminals.
>
> Please see our response to a similar question by Reviewer apti.
>
> > 520: What does it mean for state q to recognize T? How do you determine what T is?
>
> The lexing automaton recognizes the union of all the languages for each terminal. It is built via a product construction where each component automaton recognizes a specific terminal T. Thus, each final state of the product automaton corresponds to a terminal T. If multiple terminals T correspond to a single final state, one can either assign priorities or nondeterministically produce all possible tokens to support contextual lexing.
>
> > 250: When you say T can be generated along some path, do you mean a path ending in an accept state?
>
> We say T can be generated along some path from q, when $q \xrightarrow{w:T T_1 \ldots T_k}^\ast q’ \in \delta^\ast$ for some $q’$ and $T_1 \ldots T_k$.
>
> > Why is there a separation between characters and CFG terminals that is mediated by the lexer?
>
> Lexers usually operate via finite automata, which efficiently match tokens at linear time with minimal lookahead. In contrast, CFG parsers incur higher overhead. Moreover, lexers effectively handle non-grammatical constructs, such as whitespace and comments. This separation ensures that the parser grammar remains concise.
>
> > How is EOS masking handled in your method?
>
> The EOS symbol is treated as a token in the vocabulary, which generates a special end-of-parsing terminal. It behaves similarly to other tokens, except that it has transitions leading back to state $q_0$, which produce the end-of-parsing terminal in the lexing transducer.
>
> > What are the inputs/outputs of $T_{Re_{A \circ V}}$?
>
> The input alphabet is $Re_{A \circ V}$, and the output alphabet is $\Gamma$. This is similar to Figure 3, except that here the input alphabet consists of a sequence of terminals rather than characters.

---

> > ### Comment · Reviewer_nQQ5 · 2025-04-08
> >
> > Thank you for providing proofs and time/space complexity analyses as requested. I will raise my score accordingly. Unfortunately I have not had time to check all the details carefully during the rebuttal period. In the final draft of the paper, I really do recommend working on the clarity issues I raised in my review.
> >
> > > We assumed a deterministic PDA for simplicity of implementation, but determinism is not strictly necessary. To handle non-deterministic PDAs, one can track multiple non-deterministic states simultaneously (similar to Thompson’s construction for NFA determinization), compute masks separately for each state, and then take the union of these individual masks. This extension allows the algorithm to accommodate non-deterministic PDAs without affecting performance in deterministic cases. We will clarify this point in the revised version of the paper.
> >
> > I would like to push back on this claim and point out that the extension to nondeterministic PDAs would not be as easy as you say. Unlike NFAs, nondeterministic PDAs can have an infinite number of stack configurations simultaneously, and even in normal form, they can have an exponential number. You would need to use one of the dynamic programming algorithms that can simulate them in cubic time (see Butoi et al. (2022)). Wouldn't you also need to alter the structure of the (inverse) token spanner table in a significant way, similarly to the Generalized LR table in Tomita's algorithm?

---

> > > ### Author Response · Authors · 2025-04-08
> > >
> > > We will be sure to address the clarity issues raised. It is indeed true that simulating a NPDA is somewhat involved: the comparison to Thompson’s algorithm was indeed not the right analogy we intended to convey. Nevertheless, the proposed method can handle nondeterminism, though the reviewer is right that the number of possible states may, in the worst case, grow exponentially with respect to the length of the prefix.
> > >
> > > The inverse token spanner doesn’t need to be changed to support nondeterminism, since it doesn’t depend on the PDA at all, but only depends on the lexer specification and the vocabulary of LLM.
> > >
> > > The reviewer is right that achieving cubic complexity would require a dynamic-programming approach, but we believe such an approach is not suitable for efficient online processing as it would require expensive computations for each individual token (we are not aware of any GCD tool using such an approach).

---

### Official Review · Reviewer_apti · 2025-03-13

**Overall Recommendation:** 4

**Summary:**

Given a language model and a certain grammar, grammar-constrained decoding (GCD) aims to ensure the output of the language model adheres to the grammar. However, this is challenging because, typically, the token vocabulary for a language model does not align with that of the grammar. In this paper, the authors aim to overcome this challenge by suggesting a novel algorithm for GCD which involves an offline stage and an online stage, the latter being more efficient than current GCD algorithms. Through pre-processing, the online token masking process becomes much more efficient. The authors also provide GreatGramma, a tool for implementing their method.

**Claims And Evidence:**

It appears the claims are supported by clear and convincing evidence. The majority of the paper describes the new algorithm. For the online procedure, there is a bit of confusion, and a rigorous definition of the functions $A$ and $D$ would help the reader understand. For the reviewer’s understanding, is this method masking out other tokens that might also be allowed? The experiment section is strong and shows significant improvement over other methods. For the sake of completeness, it would be useful also to also report some sort of metric as regards to quality of the output using GreatGramma as well as other methods.

**Essential References Not Discussed:**

n/a

**Experimental Designs Or Analyses:**

The experimental design and analysis appears to be complete. As mentioned earlier though, the method would be even further motivated by reporting on the quality of the outputs via GreatGramma.

**Methods And Evaluation Criteria:**

The methods appear to make sense for the problem at hand. The method mostly consists of an offline processing procedure in order to create an inverse token spanner table, which is then used at inference time.

**Other Comments Or Suggestions:**

See above

**Other Strengths And Weaknesses:**

This paper appears to have a strong result. Yet, the paper is not particularly straightforward to follow. The reviewer suggests that, in order to improve clarity, the authors provide clear definitions in the beginning of sections and attempt to make Section 3 more concise and clear. For example, one point of confusion is what the “token spanner table” is, as mentioned in Figure 1. Later on, there is mention of an “inverse token spanner table.” What is the precise difference between these two, and if they are not the same, what is the point of Figure 1(d)? Furthermore, in places where definitions are provided, they are not rigorous. For example, Definition 3.2 is lacking in clarity: what are $q$ and $q’$? What is the point of the second paragraph regarding “Maximal Munch Principle” (lines 145-156)? It is unclear if the strict interpretation is being assumed or not.

**Questions For Authors:**

See above

**Relation To Broader Scientific Literature:**

Methods like verifiers and grammars are in general becoming more popular for language models. It appears as though some sort of supervision may be useful. Hence, methods to improve efficiency for such methods should allow the literature to continue to explore these directions with a more scalable approach.

**Theoretical Claims:**

Neither one of Propositions 3.4 and 3.5 have accompanying proofs. This would be useful in order to verify the claims, or to point the reader to another source proving the same result.

---

> ### Author Rebuttal · Authors · 2025-04-01
>
> We thank the reviewer for their insightful feedback. Below, we provide a detailed response to each comment and question.
>
> > is this method masking out other tokens that might also be allowed?
>
> The masking algorithm of GreatGramma is sound and complete under the assumptions on the lexer stated in the paper. We will include a proof sketch in the body of the revised paper based on the following key lemmas, and include detailed proofs in the appendix.
>
> Lemma 1. Let $T_{A \circ V} = (Q, V, \Gamma, q_0, \delta, F)$ be a token-level lexing transducer for the vocabulary $V \subseteq \Sigma^+$. Then $q_0 \xrightarrow{w:T_1 \ldots T_k}^{\ast} q' \in \delta^\ast$
> if and only if $Lex(w) = (T_1 \ldots T_k, w_r)$ for some $w_r \in \Sigma^\ast$ and $q' \in Q$ such that $q_0 \xrightarrow{w_r:\epsilon}^\ast q' \in \delta^\ast$.
>
> The proof proceeds by induction on $|w|$. The base case holds since each terminal is not nullable, and the inductive step follows directly from the construction of the lexing transducer.
>
> Lemma 2. If $q \xrightarrow{w: T_1 \ldots T_k}^\ast q'$ via the token-level lexing transducer for some incomplete token prefix $w$ of $L(G)$, then there exists a terminal $T$ producible along some path from $q'$ such that $T_1 \ldots T_k T$ is terminal prefix of $L(G)$.
>
> Proof sketch. By assumption, there exists a string $v$ such that $wv$ is in the language. Let $T_1 \ldots T_k T_{k+1} \ldots T_m $ be the lexed terminal sequence corresponding to $wv$, which is a sentence in the CFG $G$.
> Since $T_1 \ldots T_k$ itself is not a complete sentence, the sequence $T_{k+1} \ldots T_m$ must be non-empty, and in particular, terminal $T_{k+1}$ can be generated.
> Thus, we conclude that $T_1 \ldots T_k T_{k+1}$ is a terminal prefix of $L(G)$.
>
> Theorem 1. (Completeness) Given a prefix $w$ of $L(G)$, a token $v$ is not masked by Algorithm 6 (i.e., $v \in V_{allowed}$) if the concatenation $wv$ is a prefix of $L(G)$.
>
> Proof sketch. By Lemma 1, the lexing transducer produces $T_1 \ldots T_k$ and reaches a state $q'$ such that $q_0 \xrightarrow{w_r: \epsilon}^\ast q'$.
> Since $wv$ is a prefix of $L(G)$, again by Lemma 1, $q_0 \xrightarrow{wv:T_1 \ldots T_k T_{k+1} \ldots T_m} q''$ for some state $q''$, which implies $q' \xrightarrow{T_{k+1} \ldots T_m} q''$.
> Then, by Lemma 2, there exists a terminal $T$ producible along some path from $q’$ such that $T_1 \ldots T_k T_{k+1} \ldots T_m T$ is a prefix of $L(G)$.
> Therefore, we have $v \in T_{inv}(q', T_{k+1} \ldots T_m T)$, which implies $v \in V_{allowed}$.
>
> Theorem 2. (Soundness) Given a prefix $w$ of $L(G)$, a token $v$ is masked by Algorithm 6 (i.e., $v \in V_{allowed}$) if the concatenation $wv$ is not a prefix of $L(G)$.
>
> The detailed soundness proof (which we do not include here due to space limits) is similar in overall structure to the proof of Theorem 1. The proof also requires the additional fact that every sequence of terminals corresponds to the lex of a sequence of tokens in  $V^\ast$.
>
> > Neither one of Propositions 3.4 and 3.5 have accompanying proofs.
>
> Both propositions follow directly from the definition of PDAs and FSMs. We will provide formal proofs in the appendix. As a quick reference, a proof can also be found in Theorem 6.5 of [NewRef1]. Additionally, Theorem 6.17 provides intuition about how to construct an FSM that over-approximates the given PDA.
>
> [NewRef1] Hopcroft, John E. et al., "Introduction to automata theory, languages, and computation."
>
> > … What is the precise difference between these two, and if they are not the same, what is the point of Figure 1(d)?
>
> While the token spanner table maps pairs of (state, token) to terminal sequences, the inverse token spanner table reverses this mapping for faster lookup in Algorithms 5–6, mapping each (state, terminal sequence) pair to the set of tokens capable of generating that sequence in that state. The token spanner table is used to compute $Re_{A \circ V}$ and the inverse token spanner table. We will clarify this relationship in the revised paper.
>
> > Furthermore, in places where definitions are provided, they are not rigorous. For example, Definition 3.2 is lacking in clarity: what are q and q’?
>
> q and q’ are (existentially quantified) source and destination states, respectively, in the transition set of the token-level lexing transducer. We will clarify the meaning of the notation and provide more rigorous definitions in the revised paper.
>
> > What is the point of the second paragraph regarding “Maximal Munch Principle” (lines 145-156)?
>
> This paragraph introduces the informal idea behind 1-lookahead lexing. This is an additional specification (beyond just maximal munch) on how the lexer should behave. Our lexing transducer is designed to satisfy both specifications. In the revised paper, we will clarify explicitly that these specifications are our design choice. Additionally, we will provide a formal proof in the appendix to demonstrate that the transducer indeed satisfies both specifications.

---

### Official Review · Reviewer_q1Dz · 2025-03-14

**Overall Recommendation:** 4

**Summary:**

This paper describes a flexible & efficient implementation of grammar-constrained decoding for LLMs. Grammar-constrained decoding uses a (usually deterministic) CFG to constrain the output space of LLM decoding. A main challenge in apply such a constraint is the mismatch between the alphabet of the constraint (usually characters for the lexer and terminals for the parser) and the LM (subword tokens). Existing implementations either require expensive offline grammar preprocessing ([Ugare et al, 2024]) or incur a large overhead during decoding ([Willard and Louf, 2023]).

The paper proposes to use a token spanner table (Figure 1(d)) to map each subword token to possible terminal sequences with 1 lookahead from each lexer state. After converting the CFG into a PDA, this table is then further partitioned into $A(q^{\mathcal{A}}, q^{\mathcal{D}})$ (the set of terminal sequences allowed by the PDA starting at lexer state $q^{\mathcal{A}}$ and PDA state $q^{\mathcal{D}}$ under any stack configuration) and $D(q^{\mathcal{A}}, q^{\mathcal{D}})$ (the set of terminal sequences allowed by the PDA starting at lexer state $q^{\mathcal{A}}$ and PDA state $q^{\mathcal{D}}$ only under certain stack configurations), which enables faster computation of token masks at decoding time (Algorithm 6).

Benchmarks show that the proposed method is 17.71x the speed of SynCode ([Ugare et al, 2024]) during offline preprocessing and 2.73x the speed during decoding (Outline ([Willard and Louf, 2023]) while faster during offline preprocessing is inpractically slow during decoding).

[Ugare et al, 2024]: https://arxiv.org/pdf/2403.01632
[Willard and Louf, 2023]: https://arxiv.org/pdf/2307.09702

**Claims And Evidence:**

The main claims of this paper is the correctness and speed of the proposed method. The high level algorithms are correct although proofs were not given in the paper. The claim on speed is well supported by the experimental results.

**Essential References Not Discussed:**

None.

**Experimental Designs Or Analyses:**

The main experiment involves speed benchmarking for offline preprocessing and decoding. Grammars for 3 programming languages are used (Go, Python, and Java). For decoding benchmarks, a set of 5 programs for each language (367, left column) is used for computing token masks. The programs are available in the supplementary material, and they are small programs (< 100 lines). Grammar complexity is the main factor that determines the decoding speed (the grammars are all deterministic PDAs if I am not mistaken), thus the small size of the programs should not be a main problem.

**Methods And Evaluation Criteria:**

Yes.

**Other Comments Or Suggestions:**

-   Figure 2 does not appear to mark all the final states. With $q_0$ being the only final state, this FST only accepts $\epsilon$. $q_2^B$ and $q_3^C$ should also be final each with a final output arc.
-   The machine in Figure 4 is called $\mathcal{T}_{\mathcal{A} \circ \mathcal{V}}$ in most places, but other also sometimes

    $\mathcal{T}_{\mathcal{A}} \circ\mathcal{T}_V$ (Algorithm 4; 245-246, left column).
-   Algorithm 4 operates on $\mathcal{T}_{\mathcal{A} \circ \mathcal{V}}$ but in lines 257-259, left column, the text says

    $\mathcal{T}_{\mathcal{A}}$. Indeed the table can be built on

    $\mathcal{T}_{\mathcal{A}}$ alone, but the description should be more consistent.
-   (267, left column) $T_{\mathrm{inv}}(q, \alpha)$, it appears that $\alpha$ should really be $T_1 \ldots T_kT$ instead.
-   (251, right column) "we directly compose the detokenizing transducer $Re_{\mathcal{A}\circ\mathcal{V}}$": I think starting from here this subsection is trying to motivate Algorithm 5, but $Re_{\mathcal{A}\circ\mathcal{V}}$ is not the detokenizing transducer, and the composition is done implicitly in Algorithm 5.

    Later, in (256, left column), $\mathcal{T}{Re_{\mathcal{A}\circ\mathcal{V}}}$ occured without being defined. Overall, I had a really hard time understanding Section 3.3 before seeing Algorithm 5. Perhaps there is a better way to present the ideas?

**Other Strengths And Weaknesses:**

Strengths
-   Most parts of this paper is well written (except Section 3.3 which I took me quite a while to fully understand)
-   The prosposed solution are both simple and effective

Weaknesses
-   ICML may not be the best venue for this paper given there is nothing about machine learning in this paper. NLP conferences such as ACL are definitely more appropriate where a larger body of conference attendees can appreciate this paper. **However, I did not take this into consideration in my "Overall Recommendation" (i.e. I gave the rating that I would give if I were reviewing for ACL)**. The area chairs should be the judge of this.

**Questions For Authors:**

Does the PDA $\mathcal{P}$ need to be deterministic? If so, could you mention this explicitly early in the paper?

**Relation To Broader Scientific Literature:**

Most recent work on constrained-decoding using CFGs are all cited in this paper, in particular, Outline ([Willard and Louf, 2023]), XGrammar ([Dong et al, 2024]), and SynCode ([Ugare et al, 2024]). Formally, these and this paper itself are all incremental parsing methods under the hood that also involves a composition with the subword-character detokenizing FST. However, the exact techniques employed in implementing the parsing & composition have significant impacts on speed. Benchmarks show that the proposed solution in this paper is faster than SynCode and Outline during decoding (XGrammar appears to have issues that prevented the benchmarks from being conducted). Additionally this paper in my opinion presents the more elegant solution compared to the existing ones.

[Dong et al, 2024]: https://arxiv.org/pdf/2411.15100
[Ugare et al, 2024]: https://arxiv.org/pdf/2403.01632
[Willard and Louf, 2023]: https://arxiv.org/pdf/2307.09702

**Theoretical Claims:**

N/A.

---

> ### Author Rebuttal · Authors · 2025-04-01
>
> We thank the reviewer for their feedback, which will significantly improve the paper. We believe ICML is an appropriate venue as it has previously published similar work in constrained decoding (e.g., https://openreview.net/forum?id=pXaEYzrFae last year).
>
> Below, we provide a detailed response to each comment and question.
>
> > Figure 2 does not appear to mark all the final states. With being the only final state, this FST only accepts $\epsilon$. $q_2^B$ and $q_3^C$ should also be final each with a final output arc.
>
> Thanks for bringing this to our attention. $q_0$ is the only final state, but the transducer appears incorrect because we omitted the transitions for EOS (from $q_2$​ and $q_3$​ back to $q_0$​). We had initially omitted them to simplify the lexing transducer, but we now realize this omission makes the figure imprecise. We will add the necessary EOS transitions.
>
> > The machine in Figure 4 is called $T_{\mathcal{A} \circ \mathcal{V}}$ in most places, but other also sometimes $T_{\mathcal{A}} \circ T_{\mathcal{V}}$ (Algorithm 4; 245-246, left column).
>
> > (267, left column) , $T_{inv}(q, \alpha)$, it appears that $\alpha$ should really be $T_1 \ldots T_k T$ instead.
>
> Thanks, we will resolve the conflicting notation in the revised version of the paper.
>
> > Overall, I had a really hard time understanding Section 3.3 before seeing Algorithm 5. Perhaps there is a better way to present the ideas?
>
> Thanks for the feedback. The primary role of Section 3.3 is to explain how to compute the always-accepted token table $A$ and context-dependent sequence table $D$. The detokenizing transducer is mostly an implementation detail used to avoid having to iterate over terminal sequences at runtime.
>
> We will update the structure of Section 3.3 to be clearer: we will first introduce the definitions of always-accepted/rejected tokens and context-dependent terminal sequences, and then briefly mention the detokenizing transducer as an implementation detail.
>
> > Does the PDA need to be deterministic? If so, could you mention this explicitly early in the paper?
>
> We assumed a deterministic PDA for simplicity of implementation, but determinism is *not* strictly necessary. To handle non-deterministic PDAs, one can track multiple non-deterministic states simultaneously (similar to Thompson’s construction for NFA determinization), compute masks separately for each state, and then take the union of these individual masks. This extension allows the algorithm to accommodate non-deterministic PDAs without affecting performance in deterministic cases. We will clarify this point in the revised version of the paper.

---

### Decision · Program_Chairs · 2025-05-01

**Decision:**

Accept (poster)

**Comment:**

The paper proposes an efficient method for gammar-constrained decoding (GCD) that aims to facilitate structured output generation (e.g., json formats). All the reviewers appreciated that the proposed method is simple and effective, with strong empirical results. Several reviewers mentioned presentation clarity issues, and the authors were able to provide satisfactory responses during the rebuttal. Due to the unanimous positive support from the reviewers, we recommend acceptance.